

# Non-steady-state Stomatal Conductance Modeling and Its Implications: From Leaf to Ecosystem

Ke Liu[1], Yujie Wang[1], Troy Magney[2], and Christian Frankenberg[1,3]

[1]Division of Geological and Planetary Sciences, California Institute of Technology, Pasadena, California 91125, USA
[2]Department of Plant Sciences, University of California, Davis, CA 95616, USA
[3]Jet Propulsion Laboratory, California Institute of Technology, Pasadena, California 91109, USA

**Correspondence:** Ke Liu (klliu@caltech.edu) and Christian Frankenberg (cfranken@caltech.edu)

**Abstract.** Accurate and efficient modeling of stomatal conductance ($g_s$) has been a key challenge in vegetation models across scales. Current practice of most land surface models (LSMs) assumes steady-state $g_s$ and predicts stomatal responses to environmental cues as immediate jumps between stationary regimes. However, the response of stomata can be orders of magnitude slower than that of photosynthesis, and often cannot reach a steady state before the next model time-step, even on half-hourly

time scales. Here, we implemented a simple dynamic $g_s$ model in the vegetation module of an LSM developed within the Climate Modeling Alliance, and investigated the potential biases caused by the steady state assumption from leaf to canopy scales. In comparison with steady-state models, the dynamic model better predicted the coupled temporal response of photosynthesis and stomatal conductance to changes in light intensity using leaf measurements. In ecosystem flux simulations, while the impact of $g_s$ hysteresis response may not be substantial in terms of daily or monthly integrated canopy fluxes, our results sug-

gested the importance of considering this effect when quantifying fluxes in the mornings and evenings, and interpreting diurnal hysteresis patterns observed in ecosystem fluxes. Furthermore, prognostic modeling can bypass the A-C$_i$ iterations required for steady-state simulations and can be robustly run with comparable computational costs. Overall, our study highlights the implications of dynamic $g_s$ modeling in improving the accuracy and efficiency of LSMs, and for advancing our understanding of plant-environment interactions.

## 1   Introduction

Modeling stomatal conductance ($g_s$), the opening and closure of tiny pores on leaves, is one of the key elements and challenges in land surface models (LSMs). Stomata regulate the gas exchange rates of plants, allowing the uptake of $CO_2$ for photosynthetic assimilation while constraining water loss through transpiration (Berry et al., 2010; Damour et al., 2010). The behavior of stomata, especially their responses to environmental variations, plays a significant role in determining the fluxes of carbon,

water, and energy between vegetated surfaces and the atmosphere (Berry et al., 2010; Buckley, 2017). Therefore, accurate and efficient modeling of $g_s$ is important for understanding the current Earth system and projecting future changes.

Stomatal conductance has been traditionally predicted with empirical models, relating $g_s$ to photosynthesis rate and environmental cues with estimated parameters from statistical regressions (Ball et al., 1987; Leuning, 1990, 1995; Medlyn et al., 2011; Damour et al., 2010). More recently, efforts have been made to constrain stomatal behavior from the principle of optimizing



water use efficiency, i.e., the trade-offs between the carbon gain and water loss of stomatal opening (Wolf et al., 2016; Venturas et al., 2018; Wang et al., 2020). Additional understanding of stomatal response includes plant hydraulic models that consider the transport of water from soil through plants into the atmosphere (soil-plant-atmosphere continuum, SPAC) (Sperry et al., 1998, 2002; Bonan et al., 2014). However, most existing stomatal models, especially those currently used to scale from leaf to canopy level and implemented in LSMs, assume steady states (Vialet-Chabrand et al., 2017). They predict the opening and

closure of stomata in stationary regimes, modeling stomatal response to environmental variations as immediate jumps between states (Vialet-Chabrand et al., 2013).

While steady-state models assume that stomatal conductance changes instantaneously with the environment, the temporal response of stomata in reality can often be an order of magnitude slower than the biochemical response of photosynthesis (Pearcy and Seemann, 1990; Vialet-Chabrand et al., 2013), and a steady state is often not reached before the next change in

conditions (Lawson and Blatt, 2014; Vialet-Chabrand et al., 2017). This slower response of $g_s$ could further impose regulations on assimilation rate via its effects on intercellular $CO_2$ concentration ($C_i$), notably under rapidly-changing incident radiation in natural environments (Kaiser and Kappen, 2000; Vialet-Chabrand et al., 2017). The mismatch and interaction of photosynthesis and stomatal response could lead to temporal variations in water use efficiency (WUE) as well (Lawson et al., 2011; Venturas et al., 2018). These can all lead to biases and it is important to consider non-steady-state temporal responses of $g_s$ for more

accurate predictions of ecosystem fluxes. Additionally, the inclusion of this factor may also contribute to the hysteresis of plant responses and ecosystem fluxes observed in natural diurnal cycles (Vialet-Chabrand et al., 2013), e.g. evapotranspiration (ET) rates tend to be higher in the afternoon under the same incoming radiation, while canopy conductance overall decreases. These patterns have often been attributed solely to the asymmetry of meteorological variables, especially in temperature and vapor pressure deficit (Zeppel et al., 2004; Bai et al., 2015; Gimenez et al., 2019; Oogathoo et al., 2020; Lin et al., 2019).

The current practice of employing $g_s$ models that assume steady-states requires iterations to converge to stable solutions at each simulation step. At the leaf level, this typically involves two nested iteration loops, first to solve the coupled photosynthesis-stomatal conductance model for $C_i$, and then to solve the leaf energy budget for leaf temperature (Collatz et al., 1991; Bonan et al., 2018), as $g_s$ affects latent heat flux through transpiration. This approach can potentially lead to numerical issues (Sun et al., 2012) and increased computational costs, particularly when upscaling with complex canopies, where angular distribution

setups are necessary (Wang and Frankenberg, 2022). However, by utilizing prognostic updates of variables, a dynamic model could simplify simulation steps and improve computational efficiency, enabling runs at finer temporal resolutions.

Moreover, accurate parameter estimation with steady-state models (e.g. linear fitting for the slope $g_1$ in empirical models) necessitates measurements to be taken after reaching each equilibrium (Leuning, 1990; Miner et al., 2017). Depending on wide-ranging stomatal response speeds (McAusland et al., 2016), obtaining one accurate response curve could take several hours

(Liozon et al., 2000; Duarte et al., 2016). Too short of a time step could result in overestimation, underestimation, or unstable results of parameter estimates with steady-state assumptions (Xu and Baldocchi, 2003), which may often be overlooked (Miner et al., 2017). Alternatively, estimates can be approached with a prognostic model by fitting the entire response curve, where steady-state measurements are not fundamentally necessary.





Limitations of steady-state stomatal modeling have driven efforts to develop dynamic models, primarily at the leaf level
(Damour et al., 2010; Vialet-Chabrand et al., 2017). Based on observed variations of $g_s$, analytical equations of sigmoidal or
exponential response have been commonly used (Naumburg and Ellsworth, 2000; Noe and Giersch, 2004; Vialet-Chabrand
et al., 2013; Vialet-Chabrand et al., 2017; Martins et al., 2016; McAusland et al., 2016); directly adding time-dependent terms
into traditional steady-state models has also been proposed (Matthews et al., 2018). While these models have demonstrated
effective performance in reproducing leaf-level responses to light intensity in controlled conditions, the impacts of including
temporal stomatal dynamics on the simulations of larger-scale fluxes under coupled variations in the natural environment
(e.g., transpiration in the coupled diurnal cycles of radiation, temperature, and vapor pressure deficit (VPD)) have not been
investigated. This may be partly due to the parametrization and complexity of many models optimized for leaf-scale predictions
(Kirschbaum et al., 1988; Vialet-Chabrand et al., 2016), which constrains the feasibility of scaling them to the canopy level in
LSMs.

In this study, we aim to: 1) implement a simplified dynamic stomatal model in the CliMA-Land model, i.e., the land com-
ponent of a new generation Earth system model within the Climate Modeling Alliance (CliMA); 2) test model performance on
leaf-level measurements and demonstrate an alternative method of parameter estimation with the non-steady-state model in a
Bayesian nonlinear inversion framework; 3) compare simulations of the dynamic model with traditional steady-state modeling,
primarily focusing on the differences in predictions of canopy fluxes and responses to coupled environmental variations on
different time scales.

## 2 Methods and Materials

### 2.1 Model framework

#### 2.1.1 Dynamic stomatal modeling

The current steady-state modeling approach in LSMs requires convergence of nested iteration loops to solve leaf fluxes at each
time step (Figure 1a) (Bonan et al., 2018). In this study, we proposed to replace the inner loop for steady solutions of the
coupled photosynthesis-stomatal conductance ($A_n - g_s$) model with prognostic updates of $g_{sw}$ at finer time steps (Figure 1b).

At each step, instead of assuming a initial $C_i$ and iterating until convergence, our framework starts with an initial $g_{sw}$ (e.g.
for the first time step of a diurnal simulation from midnight, this can be set as the minimal conductance in dark). Then solves
$A_n$ and $C_i$ with biochemical demand and diffusive supply of internal $CO_2$ (Figure 1). For instance, when applying the Farquhar
photosynthesis model for C3 plants (Farquhar et al., 1980), with a given $g_{sw}$, the RubisCO limited rate ($A_c$) and light limited
rate $A_j$ are calculated using:

$$A_c = V_{cmax} \cdot \frac{C_i - \Gamma^*}{C_i + K_m} = g_{lc} \cdot (C_a - C_i) + R_d, \tag{1}$$

$$A_j = J \cdot \frac{C_i - \Gamma^*}{4C_i + 8\Gamma^*} = g_{lc} \cdot (C_a - C_i) + R_d. \tag{2}$$





where the middle parts in Eq. 1 and Eq. 2 represent the biochemical demand, and the right part represents the diffusive

supply limitation of photosynthesis. $V_{\text{cmax}}$ is the maximum carboxylation rate, $C_a$ is the ambient $CO_2$ concentration, $R_d$ is the

respiration rate, $\Gamma^*$ is the $CO_2$ compensation point with the absence of respiration, $J$ is the electron transport rate, $K_m$ is the

Michaelis-Menten's coefficient, $g_{\text{lc}}$ is the leaf total conductance to $CO_2$, which can be calculated using: $g_{\text{lc}}^{-1} = g_{\text{bc}}^{-1} + 1.6 g_{\text{sw}}^{-1} +$

$g_m^{-1}$, with $g_{\text{bc}}$ the boundary conductance to $CO_2$ and $g_m$ the mesophyll conductance. Note that computing $A_c$ or $A_j$ requires

solving for $C_i$ first. With a known $g_{\text{lc}}$ from $g_{\text{sw}}$ at each time-step, rearranging Eq. 1 and Eq. 2 allows for the analytical solution

of $C_i$, $A_c$ and $A_j$, respectively.

For prognostic updates of $g_{\text{sw}}$, we implemented a simplified dynamic model, adapted from previous studies on leaf-level

prognostic modeling (Kirschbaum et al., 1988; Rayment et al., 2000; Noe and Giersch, 2004; Vialet-Chabrand et al., 2016):

$$\frac{\Delta g_t}{\Delta t} = \frac{(g_{\text{ss}} - g_t)}{\tau} \tag{3}$$

where $\Delta t$ is the time step of the simulation, $g_t$ represents the conductance at the current time step, $g_{\text{ss}}$ is the target conductance

calculated with steady-state models at the current conditions, and $\tau$ is the time constant, representing the time scale of stomatal

responses.

As indicated in the flow chart (Figure 1), our dynamic modeling avoids nested iterations for steady solutions while requires

updates of variables at finer time steps (e.g. 5-10 min, compared to 30 or 60 min time steps of current LSMs) for the stability

of simulations, which we tested and discussed in Section 2.3.3 and 3.3. The prognostic updates of leaf temperature can be

implemented accordingly, but as it is not within the scope of this study, we prescribed the leaf temperature updates with

measurements in our simulations.

### 2.1.2  Implementation in LSM

CliMA Land (https://github.com/CliMA/Land), a new generation LSM, is highly modularized and offers flexible model schemes

(Wang et al., 2021, 2023), enabling easy implementation and assessment of the dynamic stomatal model across scales. In this

study, we used the classic photosynthesis model developed by Farquhar et al. (1980) for C3 plants. For stomatal conductance,

we implemented the non-steady-state modeling framework in CliMA Land, and the steady-state $g_s$ responses were predicted

using the Ball-Berry model (Ball et al., 1987) and the Medlyn model (Medlyn et al., 2011).

### 2.2  Performance on leaf level measurements

### 2.2.1  Leaf gas exchange

To test our model and determine key parameters, we recorded light response curves of grape (*Vitis vinifera*) and walnut (*Juglans

*regia cv.*) leaves using a LI-6800 portable photosynthesis system (LI-COR, Inc., Lincoln, NE, USA). Saplings of *Vitis vinifera*

and *Juglans regia cv.* were planted in 5-gallon pots with UC soil mix. 44.4 mL of Osmocote® Smart-Release® Plant Food

Plus fertilizer were added to each pot. The plants were grown in a UC Davis lath house. The plants were watered to maintain

around 75 percent of completely saturated soil by weight (details in Meeker et al. (2021)). The youngest, fully expanded, intact





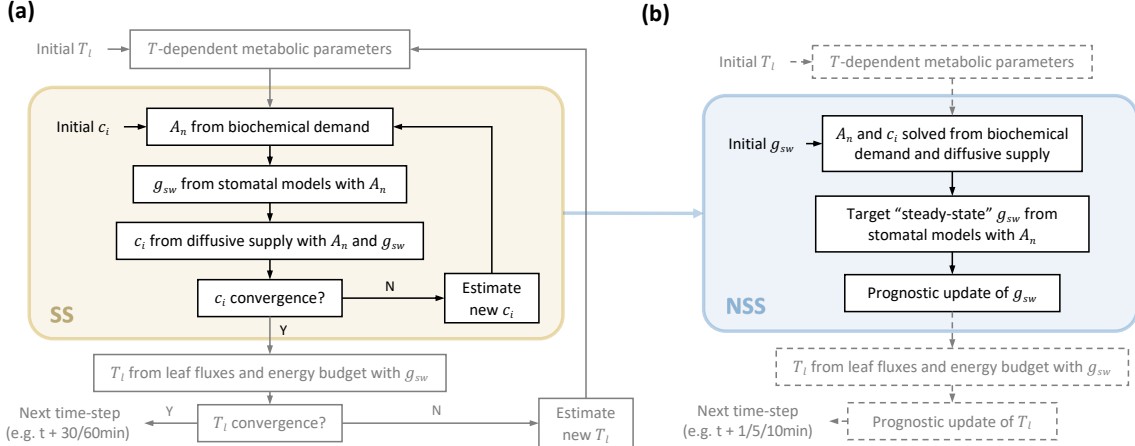

**Figure 1.** Comparison of leaf flux calculation flows in (a) steady-state (SS) and (b) non-steady-state (NSS) dynamic modeling. (a) illustrates the two nested loops at each time step in the current practice of steady-state framework, adapted from Bonan et al. (2018). The inner iteration in the light yellow box represents the flow of solving the coupled photosynthesis-stomatal conductance ($A_n$-$g_s$) model for $C_i$. The outer solves the leaf energy budget for leaf temperature ($T_l$). The focus of this study is to implement and compare a dynamic modeling framework to the $A_n$-$g_s$ model, illustrated in light blue box in (b), where, instead of iterating for steady solutions, $g_{sw}$ is updated prognostically at finer time steps, based on environmental conditions and a simplified dynamic model (Section 2.1.1). This NSS framework of modeling $g_{sw}$ also allows prognostic updates of $T_l$. As its implementation is not within the scope of this study, related flows are shown in dashed parts.

leaf was chosen and dark-adapted for 30 min. During the measurements, the photosynthetic photon flux density (PPFD) was sequentially increased following the gradient of 50, 100, 200, 400, 600, 900, 1200, 1500, 1800 $\mu$mol m$^{-2}$ s$^{-1}$, with a time step of 30 min at each light level. The chamber air temperature was set at 25 °C; CO2 partial pressure was controlled at 400 ppm; the relative humidity in the chamber was maintained around 50%.

### 2.2.2 Parameter optimization

We applied a Bayesian nonlinear inversion framework (Rodgers, 2000; Dutta et al., 2019) to jointly fit the response curves of the net photosynthetic assimilation ($A_n$) and stomatal conductance ($g_s$) for each leaf with the non-steady-state model. The forward problem in this case can be represented as follows:

$$y = \mathcal{F}(X; b) + \epsilon; \tag{4}$$

where $y$ represent the measurements, i.e. the light response curves of both $A_n$ and $g_s$ (see Section 2.2.1); $\mathcal{F}$ represents the
forward model, CliMA-Land with the dynamic $g_s$ model (see Section 2.1); $X$ is the state vector of parameters to be retrieved, which in our case includes: the maximum carboxylation rate ($V_{cmax}$), the slope ($g_1$) and the minimum conductance ($g_0$) of the BB model, the mesophlly conductance ($g_m$) (Sun et al., 2014), and the time constant ($\tau$). We also included a scaling factor for $A_n$, to account for variations in the respiration rate and the ratios between $CO_2$ and $H_2O$ fluxes; $b$ is the vector of other



parameters that have influences the measurements, are known to some accuracy but not intended to be retrieved, e.g. the ratio
between $J_{max}$ (the maximum electron transport rate) and $V_{cmax}$, which is assumed to be 1.6 in this study but may vary across
conditions (Medlyn et al., 2002); and $\epsilon$ is the error term.

The Levenberg–Marquardt (LM) iterations (Levenberg, 1944; Marquardt, 1963; Rodgers, 2000) were utilized to solve the
nonlinear inversion problem and find the best estimate of key parameters:

$$x_{i+1} = x_i + \left((1+\gamma)S_{\mathrm{a}}^{-1} + K_i^T S_\epsilon^{-1} K_i\right)^{-1} \left(K_i^T S_\epsilon^{-1}[y - F(X_i)] - S_{\mathrm{a}}^{-1}[x_i - x_{\mathrm{a}}]\right) \tag{5}$$

where $x_{\mathrm{a}}$ is the prior estimate of the state (in this study, $V_{cmax}$: 70 mol m$^{-2}$s$^{-1}$, $g_1$: 9, $g_0$: 0.03 mol H$_2$O m$^{-2}$ s$^{-1}$, $g_{\mathrm{m}}$: 0.4
mol CO$_2$ m$^{-2}$ s$^{-1}$, $\tau$: 600 s, $scaling_{\mathrm{A}}$: 1); $S_{\mathrm{a}}$ is the prior covariance matrix, assumed to be purely diagonal, with Gaussian
uncertainties in the prior state (the assumed prior standard deviation of $V_{cmax}$: 30, $g_1$: 3, $g_0$: 0.005, $g_{\mathrm{m}}$: 0.02, $\tau$: 100, $scaling_{\mathrm{A}}$,
0.01). $K_i$ is the Jacobian matrix at the $i$th iteration. $\gamma$ is adjusted at each step, ensuring that each update of the state vector
moves towards minimizing the cost function. $S_\epsilon$ is the error covariance matrix; in our case, errors were assumed to be mainly
from measurement uncertainties and calculated based on the standard deviation and mean of the $\Delta CO_2$ and $\Delta H_2O$ in LI-6800
measurements.

### 2.2.3 Uncertainties in traditional parameter estimation

To illustrate the influence of time steps on parameter estimation in the traditional method, which assumes steady states, we used
the NSS model to generate leaf response curves to the same PPFD sequence but with different time intervals. For example,
in the 5-min time step simulation, light intensity input jumped every 5 minutes, and measurements were assumed to be taken
right before the next jump, following the traditional method. We then employed these curves to calculate the estimated $g_1$ and
$g_0$ values using the traditional linear fitting method for the Ball-Berry model. The potential biases were assessed by comparing
fitted parameters with different applied time steps.

## 2.3 Comparison of models in diurnal cycles

To compare the prediction of surface flux from models with different assumptions and to assess the potential bias of steady-
state modeling, we employed high temporal resolution radiation measurements in the field as inputs and ran CliMA Land with
both setups. We evaluated and compared the simulation results on both the leaf and canopy flux scales.

### 2.3.1 Diurnal variations of radiations

Photosynthetically active radiation (PAR) in a crop field (42.481677°N, 93.523521°W) was recorded with a LI-190R quantum
sensor (LI-COR, Inc., Lincoln, NE, USA) at 1 s temporal resolution during August 2017.

In addition to the fluctuations of total incoming photon density that the PAR sensor can provide, canopies in natural environ-
ments also experience variations in the fraction of direct and diffuse components in the total radiation. This variation affects
the distribution of PAR received by individual leaves across different layers of the canopy structure (Durand et al., 2021). To
account for this effect, we employed an empirical fitting with the hourly radiation data from ERA5 (Figure 2), to estimate the





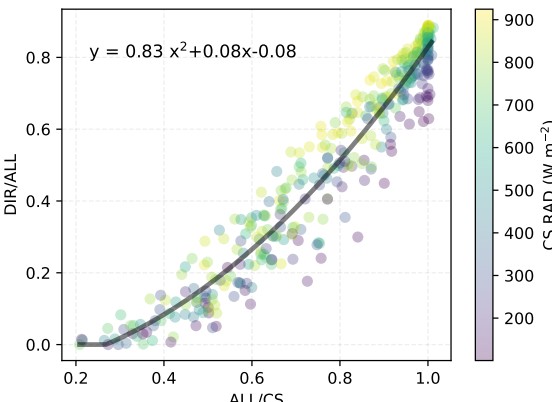

**Figure 2.** The empirical relationship between the direct radiation (DIR) fraction and the ratio between the total downward solar radiation (ALL) and the clear sky radiation (CS) in August 2017, filtered by 100 W m$^{-2}$ CS radiation. Radiations are from the ERA5-Land hourly dataset.

partitioning between the direct and diffuse radiation (Boland et al., 2001). The empirical relationship was then applied to high temporal resolution PAR measurements to obtain the direct and diffuse components in the recorded total radiation, which were used as inputs for simulations at the canopy scale.

### 2.3.2  Environmental drivers and plant traits

Meteorological variables (e.g. air temperature, dew-point temperature, volumetric soil water, wind speed etc.) from the ERA5-
Land hourly dataset were input as environmental drivers for the simulations on the canopy scale. Linear interpolations were applied for runs at sub-hourly time steps. Key plant traits (e.g. $V_{cmax}$, $g_1$, leaf area index (LAI)) were extracted from several globally gridded datasets using GriddingMachine (Wang et al. 2022; Croft et al. 2020; Butler et al. 2017; Luo et al. 2021; De Kauwe et al. 2015; Yuan et al. 2011; He et al. 2012, ; also see Wang et al. (2023) for detailed information on global scale datasets used in CliMA-Land).

### 2.3.3  Model simulations

In the comparison of the dynamic (non-steady-state, NSS) and steady-state (SS) modeling, a time constant of 900 s was used for the prognostic model, based on the average time constant retrieved from leaf response curve in Section 2.2.1 and 2.2.2 as well as previous studies on the time constant variations (Vialet-Chabrand et al., 2013; McAusland et al., 2016; Vialet-Chabrand et al., 2017). In the SS runs, iterations were employed to converge to steady-state solutions at each time step.

For the leaf-scale runs, we used the key parameters retrieved in previous sections and tested model predictions for an ideal clear-sky day. To investigate the differences in ecosystem fluxes, we ran and assessed the NSS and SS simulations using a time step of 1 minute, with the inputs of meteorological drivers and plant traits at the location of the PAR measurement for




the month of August 2017. In order to further evaluate the potential contribution of $g_s$ hysteresis to the observed diurnal hysteresis of ecosystem fluxes, we compared the standard runs with the model predictions where environmental variables (e.g. temperature, VPD, soil water content (SWC), etc.) were held constant over the daytime (as the mean of daytime values in each day). This approach allowed us to isolate the effect of hysteresis in $g_s$ response and assess its potential contribution to the observed diurnal hysteresis of canopy and ecosystem fluxes.

Furthermore, to test the stability of prognostic modeling and assess the computational cost, we compared NSS simulations using different time steps, as well as the SS simulation run at a time step of 30 min, which is commonly used in current LSMs. This enabled us to evaluate the sensitivity of NSS predictions to the time step used, as well as compare the computational cost for stable NSS runs and standard SS simulations. We resampled the environmental drivers from ERA5 and the PAR sensor to match the temporal resolution of the simulations, while maintaining constant average values for each diurnal cycle across simulations with different time steps.

## 3 Results

### 3.1 Model performance and parameter estimates on leaf measurements

With the parameters estimated from the LM inversion framework, the non-steady-state model well predicted the temporal responses of $g_{sw}$ and $A_n$ (Figure 3). The model was able to capture the gradual increases of $g_{sw}$ and $A_n$ after each step change in $APAR$, and the reproduced curves were close to the measurements, with all $R^2$ higher than 0.98. Fitted time constant $\tau$ showed a variation across leaves (292 s and 2028 s for the *Vitis vinifera* leaf and the *Juglans regia cv.* leaf, respectively). The relative difference in the time constant matched with the variations of response speed observed in the measured response curves (Figure 3). Compared to the SS model, the dynamic model provided more accurate prediction to the temporal responses. The improvements in $R^2$ were more prominent in the predictions of the *Juglans regia cv.* leaf responses, which have a larger time constant, than in those of the *Vitis vinifera* leaf. Other parameters estimated for the *Vitis vinifera* leaf include $V_{cmax}$, 71 mol m$^{-2}$s$^{-1}$, $g_1$, 11.3, $g_0$, 0.023 mol H$_2$O m$^{-2}$ s$^{-1}$, $g_m$, 0.18 mol CO$_2$ m$^{-2}$ s$^{-1}$, $scaling_A$, 1.1; for the *Juglans regia cv.* leaf, $V_{cmax}$, 152 mol m$^{-2}$s$^{-1}$, $g_1$, 3.9, $g_0$, 0.052 mol H$_2$O m$^{-2}$ s$^{-1}$, $g_m$, 0.34 mol CO$_2$ m$^{-2}$ s$^{-1}$, $scaling_A$, 1.0.

The dynamic model was also able to better capture the temporal variations of internal CO$_2$ concentration (Figure 4). Particularly, the NSS model reproduced the undershooting of the intercellular CO$_2$ concentration ($C_i$) after each step change in light intensity, which resulted from the differences in the speed of $g_s$ and $A_n$ responses and their interactions. As shown in the measured time series (Figure 3), after each increase in the incident light, photosynthesis was able to respond almost instantaneously, leading to a rapid decrease in $C_i$, while stomata opened gradually, slowly bringing up $C_i$ over time. This then led to a gradual rise of $A_n$ after the initial rapid response, indicating the regulation of $g_s$ on $A_n$ through its impacts on the internal CO$_2$ supply. In the meantime, the increasing $A_n$ further promoted the opening of stomata with a higher internal CO$_2$ demand, demonstrating their coupled responses to environmental variations.

With the dynamic model and optimized parameters that accurately reproduced the measured leaf responses, we investigated the influence of time steps used in light response curves (i.e. the length of intervals between step changes in light intensity) on



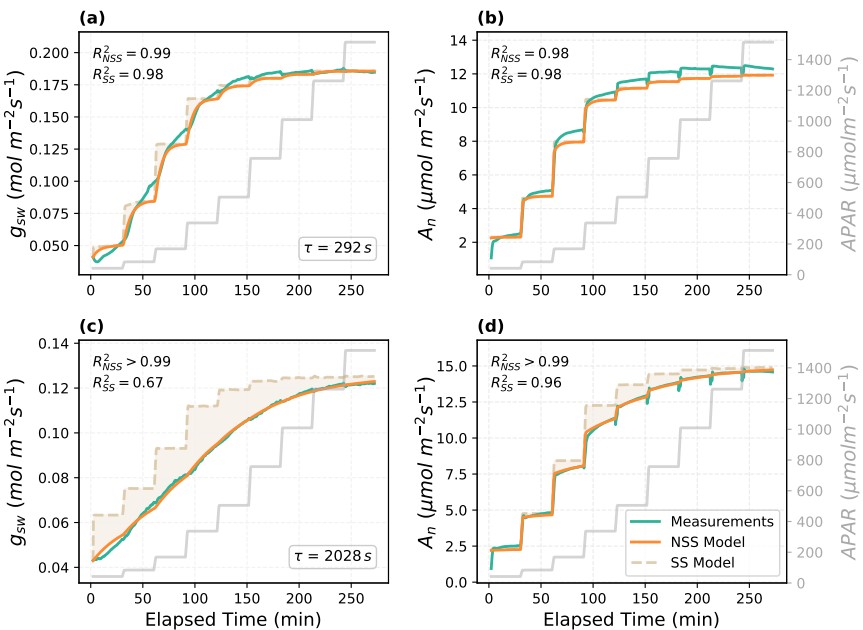

**Figure 3.** Modeled and measured temporal responses of the stomatal conductance ($g_{sw}$) and net photosynthesis rate ($A_n$) to the step changes in APAR for different leaves. The shaded area indicates the difference between the prediction of the steady-state (SS) model and the non-steady-state (NSS) dynamic model. (a-b) The temporal responses of the *Vitis vinifera* leaf, (c-d) the *Juglans regia cv.* leaf.

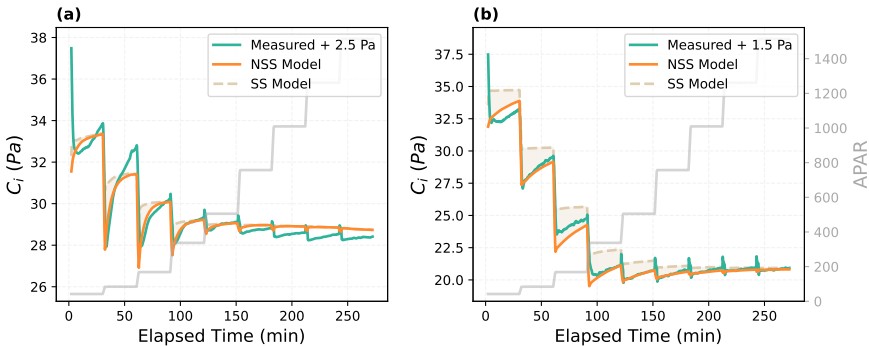

**Figure 4.** Modeled and measured temporal responses of intercelluar $CO_2$ concentration ($C_i$) for (a) the *Vitis vinifera* leaf and (b) the *Juglans regia cv.* leaf. As indicated in labels, measured curves were shifted to illustrate the comparison of modelled and measured response patterns, as the absolute values are not directly comparable due to different assumptions of LI-6800 and CliMA-Land in calculating the internal $CO_2$. The shaded area indicates the difference between the prediction of the steady-state (SS) model and the non-steady-state (NSS) dynamic model.



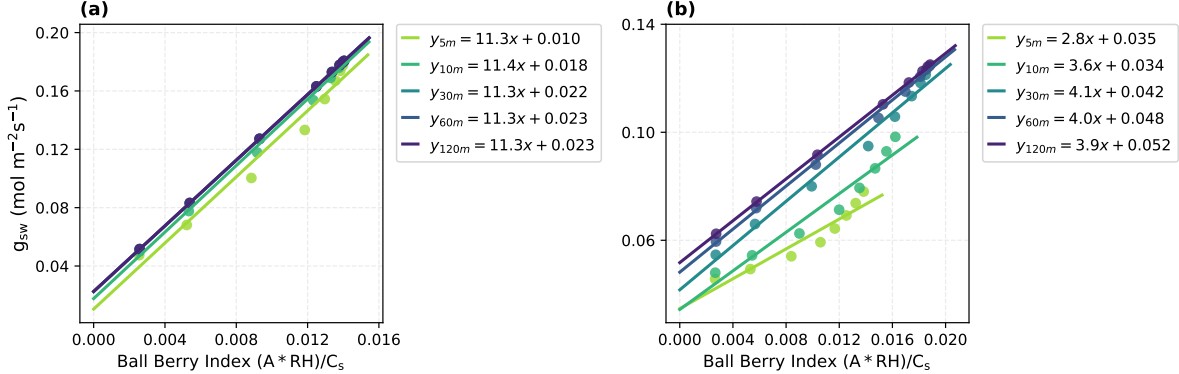

**Figure 5.** Parameter estimates for the Ball-Berry model with the traditional linear fitting method using model-reproduced response curves with different time steps (5 min, 10 min, 30 min, 60 min, 120 min). (a) fitting results for the *Vitis vinifera* leaf, (b) the *Juglans regia cv.* leaf. Corresponding Ball-Berry index and $g_{sw}$ are plotted, along with the fitted lines and parameters (i.e. the Ball-Berry slope, $g_1$, and the intercept, namely, the minimum conductance, $g_0$).

parameter estimates obtained with traditional methods (Figure 5). The results showed that, particularly for the *Juglans regia cv.* leaf that has a long time constant over 2000 s, the values and relationship between the Ball-Berry index and $g_{sw}$ varied significantly depending on the time step used, resulting in notable uncertainties in fitted $g_1$ and $g_0$ with too short of a interval to reach equilibrium. This also suggested that obtaining reliable estimates for this leaf using the traditional method could require more than an hour for stable readings at each step.

## 3.2 Model comparison in diurnal cycles

### 3.2.1 Leaf responses

To compare NSS and SS models over the course of a day, we evaluated the differences in their predictions of leaf responses to an ideal diurnal cycle of light with other environmental conditions (e.g. temperature, VPD, $CO_2$) held constant (Figure 6). Results showed that compared to NSS, the SS model predicted a higher $A_n$ and $g_s$ in the morning, as it assumed the stomata could respond immediately to an increase in light, whilst in the more realistic NSS simulation, the gradual opening of stomata limited the $CO_2$ supply for photosynthesis with a lower $C_i$. The opposite was true for the afternoon, but the overestimation of $A_n$ and $g_s$ in SS modeling in the morning was more significant than the underestimation in the afternoon, leading to slightly higher diurnally-integrated predictions than those of the NSS model. This was due to the fact that in the course of sunset, the major limiting factor on productivity was the decreasing light, in contrast to the sunrise where it was the available $C_i$ regulated by $g_s$ responses that mainly constrained $A_n$ increases. The relative differences (RDs) in integrated $g_s$ in the morning and afternoon were both higher than those of the photosynthesis, reflecting the differences in the response speed.



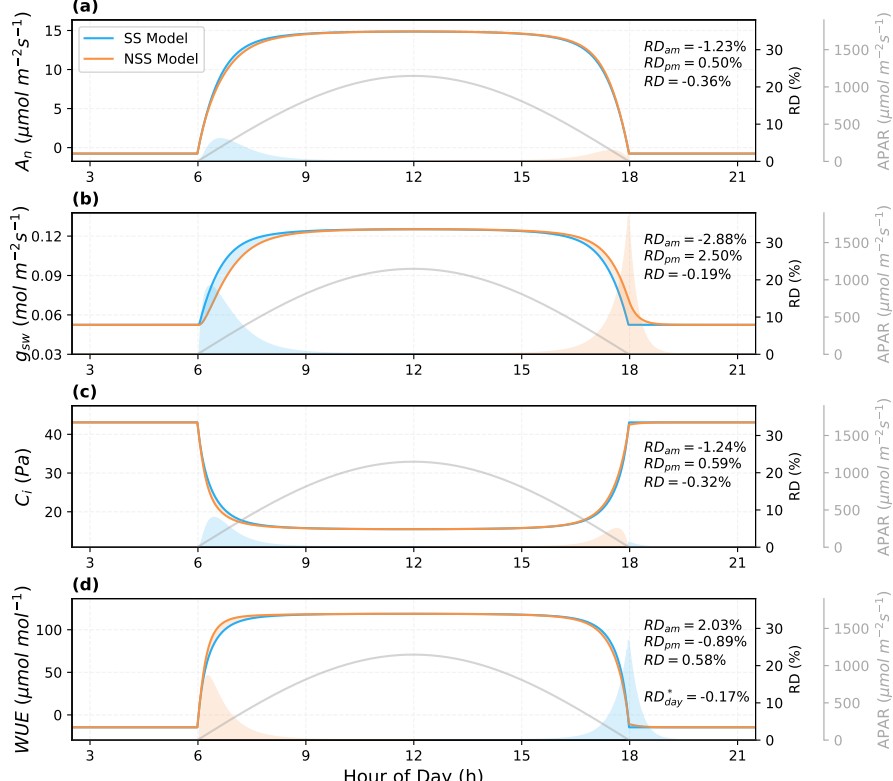

**Figure 6.** Predictions of the leaf diurnal course of (a) net photosynthesis rate ($A_n$), (b) stomatal conductance to water vapor ($g_{sw}$), (c) intercellular $CO_2$ concentration ($C_i$), and (d) intrinsic water-use efficiency (WUE) for a leaf with a stomatal time constant of 900 s in an ideal clear-sky day. Other environmental conditions (e.g. leaf temperature, VPD) were held constant. The shaded areas indicate the differences between the NSS and SS simulations (blue: SS > NSS; orange: SS < NSS), both in absolute and relative terms. Relative differences (RD, NSS - SS) in the temporal integrals are also presented, for morning (am, 5:00-12:00), afternoon (pm, 12:00-19:00), and daytime (5:00-19:00). RD* of WUE represents the ratio between integrated $A_n$ and $g_{sw}$, differing from the RD, the integral of the instantaneous WUE during the daytime.

The differences in predictions of $A_n$ and $g_{sw}$ responses also led to RDs in the intrinsic water-use efficiency (WUE, i.e. the ratio between $A_n$ and $g_{sw}$). Although the mean instantaneous WUE during the daytime was higher in the NSS simulation, diurnal WUE calculated from the integrated $A_n$ and $g_{sw}$ was lower. This was because the gradual opening of stomata during the sunrise limited assimilation in the morning, whereas during the sunset, delayed closure led to unnecessary water loss when carbon gain was constrained by low light.




### 3.2.2 Canopy fluxes

To quantify the impacts of the inclusion of $g_s$ temporal response, we analyzed the simulated canopy fluxes under natural
radiation variations and coupled dynamics of environmental conditions. As shown in the examples of diurnal cycle simulations
(Figure 7 and 8), the SS model predicted higher variations in instantaneous fluxes in response to rapid fluctuations in radiation,
particularly in transpiration rates.

The differences in fluxes between the NSS and SS predictions were not significant when integrated over monthly periods
(e.g. the mean RD of transpiration in August 2017, 0.87 %, and the median, 1.0 %), but are notable in diurnal cycles depending
on the radiation and other conditions, especially when considering the sub-diurnal scale (e.g. the maximum RD in transpiration
for daytime integrals was 2.74 % and the variation of afternoon RDs ranged from -7.4 % to 6.1 %).

The overall tendency to overestimate productivity with traditional SS models was also observed on the canopy scale, as the
regulation of $g_s$ hysteresis on the supply of $CO_2$ for photosynthesis was not considered (Figure 9b). For example, in Figure 7,
when rapid spikes of radiation occurred in the afternoon, the speed of $g_s$ response constrained the increases of photosynthesis
in the NSS simulation.

In contrast to the leaf-scale results, when accounting for other co-varying environmental drivers (e.g. temperature, VPD, soil
water content), the SS model tended to underestimate canopy transpiration rates, although RDs in the mornings and afternoons
varied depending on the radiation dynamics (Figure 7b, Figure 8b, Figure 9a). This could be because the transpiration rates
were determined by both $g_{sw}$ and VPD. During the daytime, VPD usually increased following the air temperature and peaked in
the afternoon, when the slow response of stomata to the increasing VPD and decreasing radiation could result in excess water
loss (Figure 8b, Figure 9 a and c). The overestimation of productivity and underestimation of transpiration in SS simulations
also led to further overestimation of the WUE.

### 3.2.3 Diurnal hysteresis

To investigate the relative contributions of $g_s$ hysteresis and environmental variables to the hysteresis observed in plant be-
haviors and ecosystem fluxes, we separated the effects of these two factors by comparing predicted response curves in NSS
and SS simulations with and without diurnal environmental variations (e.g. temperature, VPD, soil water content). While the
asymmetry of environmental variables in the diurnal cycle could lead to a modeled hysteresis of $g_s$ in response to radiation,
where $g_s$ tended to be lower in the afternoon mainly due to higher VPD and temperature, our results (Figure 10) showed that
the kinetic lag of $g_s$ could partially offset this effect (Figure 10 b and d), even presenting an opposite tendency at low radiations.
Additionally, only the NSS model simulations predicted a hysteresis of canopy transpiration, with or without the consideration
of coupled environmental variations (Figure 10 g and h), in which canopy $H_2O$ fluxes tended to be higher in the afternoon.



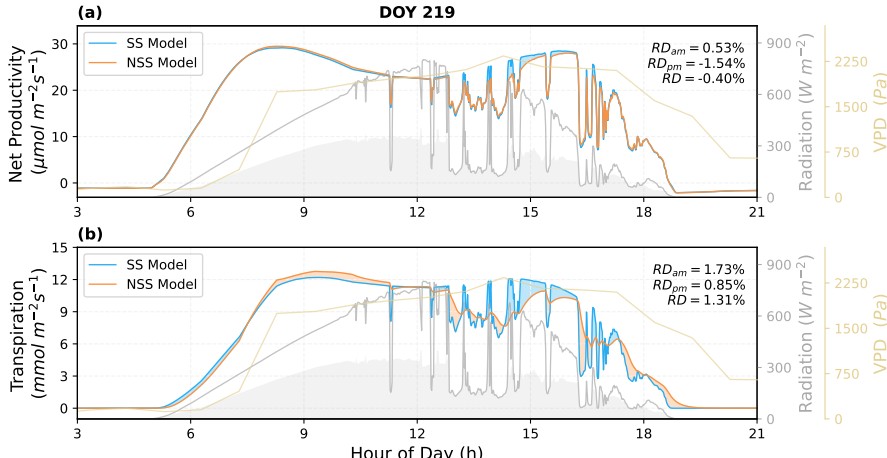

**Figure 7.** Comparison of the predicted diurnal cycles of ecosystem fluxes, (a) the net productivity and (b) the transpiration rate for DOY 219, 2017. The shaded areas in blue and orange indicate the differences between the NSS and SS predictions (blue: SS > NSS; orange: SS < NSS). The shaded areas in gray under the radiation curves represent the diffuse component of the total radiation. Relative differences (RD, NSS - SS) in the temporal integrated fluxes are also presented, for morning (am, 5:00-12:00), afternoon (pm, 12:00-19:00), and daytime (5:00-19:00).

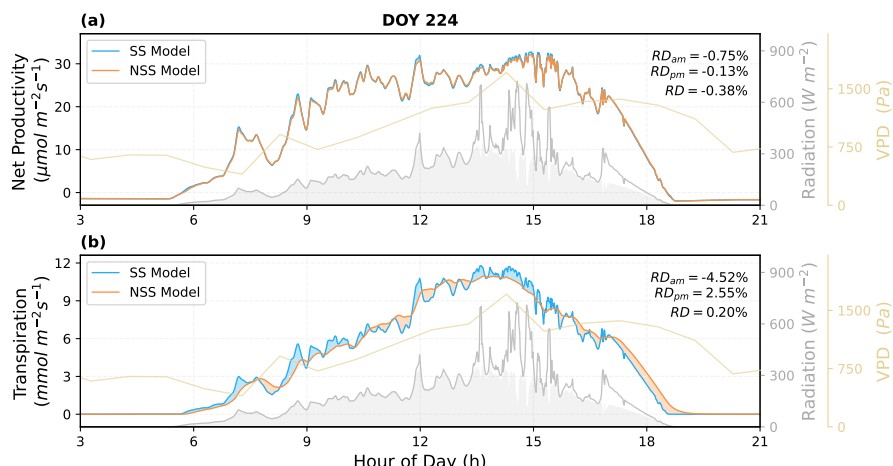

**Figure 8.** Comparison of the predicted diurnal cycles of ecosystem fluxes, (a) the net productivity and (b) the transpiration rate for DOY 224, 2017. The shaded areas in blue and orange indicate the differences between the NSS and SS predictions (blue: SS > NSS; orange: SS < NSS). The shaded areas in gray under the radiation curves represent the diffuse component of the total radiation. Relative differences (RD, NSS - SS) in the temporal integrated fluxes are also presented, for morning (am, 5:00-12:00), afternoon (pm, 12:00-19:00), and daytime (5:00-19:00).





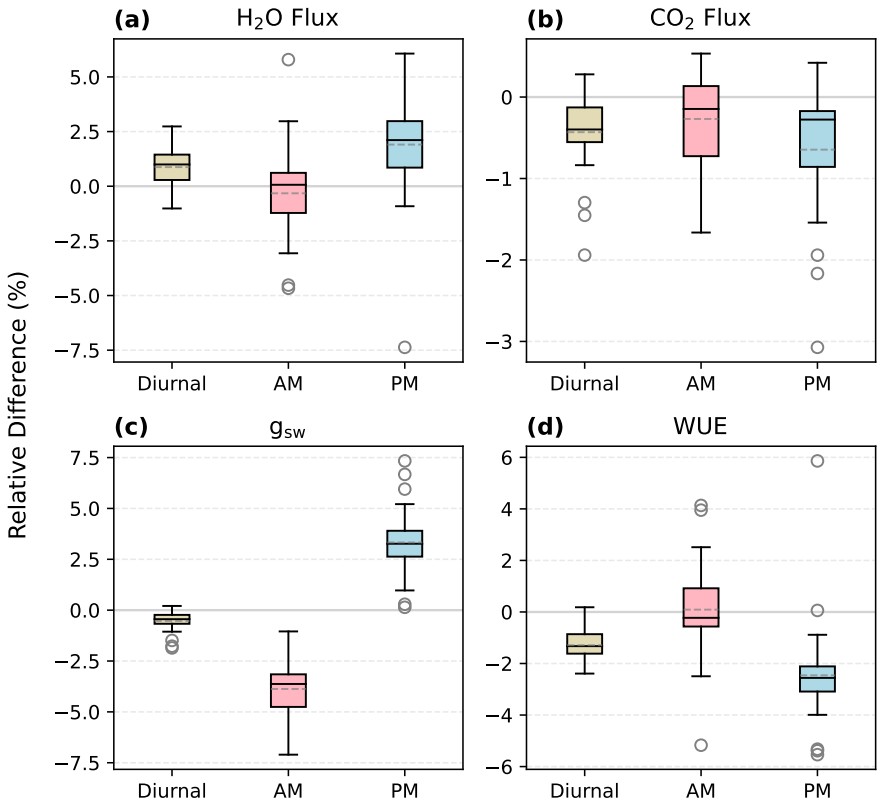

**Figure 9.** Relative differences (NSS - SS) in the predicted daytime-mean fluxes of the NSS (time step: 1 min) and SS (1 min) simulations for August 2017. The solid line in each box indicates the median, and the dashed line represents the mean. (a) RDs in $H_2O$ flux, the transpiration rate, (b) $CO_2$ flux, the net productivity, (c) Canopy-averaged stomatal conductance to water ($g_{sw}$), (d) Water-use efficiency (WUE). Diurnal: 5:00-19:00, AM: 5:00-12:00, PM: 12:00-19:00.

### 3.3 Stability of the dynamic model

We further assessed the sensitivity of the dynamic modeling to the time step of simulation. Figure 11 shows the NSS model was be able to run at a time step of 10 minutes stably and still demonstrated the impacts of gradual $g_s$ responses, as compared to the traditional practice of SS modeling at a time step of 30 minutes.

### 4 Discussion

In this study, we demonstrated the feasibility and benefits of implementing a non-steady-state stomatal conductance modeling framework from the leaf to canopy scale, in a new generation LSM, CliMA-Land. Our results suggested that compared to traditional steady-state models, the dynamic model was able to provide more realistic and accurate predictions of leaf temporal





responses to the changes in light intensity (Section 3.1). In the meantime, modeling $g_s$ with prognostic updates - similar to how plants control their stomata movements gradually in natural environments - neither increased computational cost nor model complexity, as simulations were simplified with iterations to solve for steady states avoided. Sun et al. (2012) pointed out the default 3-step fix-point iteration in CLM4 (the Community Land Model version 4) does not always converge, leading to uncertainties in flux predictions. In our simulations at the canopy scale (Section 3.3), the dynamic model could be stably run

at a temporal resolution that presented comparable efficiency to the current practice of 30-minute resolution SS simulations commonly used in LSMs. This also indicates the dynamic model can enable predictions of canopy flux dynamics at a finer time resolution with higher efficiency and accuracy.

  With the non-steady-state model, we were able to apply a Bayesian nonlinear inversion framework to jointly fit the light response curves of both $A_n$ and $g_s$, and obtain estimates for key parameters (Section 3.1). As suggested in our results (Figure 5)

and previous studies (Xu and Baldocchi, 2003; Miner et al., 2017), the time step of light response curves can notably influence the estimated parameters obtained from the traditional linear fitting method for steady-state empirical models. Our framework with the dynamic model can help reduce the time required for accurate parameter estimations, particularly for leaves with long time constants, as equilibrium is not required. Although the retrieval setups presented in this study may not be optimal for estimating $V_{cmax}$, which is typically derived from $A$-$C_i$ response curves (Medlyn et al., 2002; Miao et al., 2009; Duarte et al.,

2016), a similar framework can be applied to other scenarios for estimation of various parameters, including a $A$-$C_i$ curve for $V_{cmax}$.

  Furthermore, we evaluated how the inclusion of $g_s$ temporal responses could affect model predictions of leaf and canopy fluxes in diurnal cycles with natural environmental variations (Section 3.2.2). The comparison of NSS and SS simulations indicated that, while the differences in monthly fluxes were not significant, effects of $g_s$ temporal dynamics could be notable

in diurnal courses and on sub-diurnal scales depending on the conditions. Overall, slow opening of stomata tended to limit productivity responses to rapid radiation increases, and delayed closure of $g_s$ following decreases in radiation or increases in environmental stress (e.g. increasing VPD), results in unnecessary water loss. Both aspects led to overestimation of canopy WUE in traditional steady-state simulations. Similar effects have been noted in studies on leaf-scale response to PPFD fluctuations (Lawson et al., 2011; Lawson and Blatt, 2014; McAusland et al., 2016). This suggests that the temporal hysteresis of $g_s$

can have impacts on the integrated cost (water loss) and benefit (carbon gain) of stomatal behavior in diurnal cycles, especially in highly fluctuating environments (e.g. understory plants experiencing sunflecks), and it is necessary to include its effects on the optimal trade-off to understand and predict stomatal responses with the optimization theory (Cowan and Farquhar, 1977; Vialet-Chabrand et al., 2017).

  In addition, the hysteresis of leaf-level $g_s$ response can contribute to the hysteresis patterns at the ecosystem scale, which

have often been solely attributed to the asymmetry of environmental variables during the daytime. For instance, higher evapotranspiration (ET) fluxes and sap velocity (i.e. an indicator of plant transpiration rate) have been observed in the field, with explanation often focused on higher VPD in the afternoon following increased air temperature (Zeppel et al., 2004; Gimenez et al., 2019; Oogathoo et al., 2020; Lin et al., 2019). Our simulations showed that the SS model with diurnal environmental variations was unable to reproduce this hysteresis pattern, while it was captured in NSS runs, indicating the significance of





considering $g_s$ temporal dynamics when interpreting diurnal hysteresis in transpiration (Section 3.2.3). Moreover, observed patterns of lower $g_s$ in the afternoon have also been commonly explained with similar environmental asymmetry (Bai et al., 2015; Lin et al., 2019), whilst our results suggested the kinetic lag of $g_s$ could partially offset this effect, and thus should be taken into account in understanding the hysteresis patterns.

Further improvements can be made in applying separate time constants for stomata opening and closing course, as previous
studies have suggested different response speeds of increasing and decreasing $g_s$ to a step change in irradiance (McAusland et al., 2016; Vialet-Chabrand et al., 2017; Matthews et al., 2018). More comprehensive measurements of temporal response curves across species and conditions can also contribute to improving our understanding of the variation of time constant towards better predictions of $g_s$ temporal responses.

The dynamic $g_s$ model enables predictions of temporal changes in latent heat flux through transpiration in leaf energy
balance, which allows a similar prognostic framework to be employed for the modeling of leaf temperature. Bonan et al. (2018) implemented a non-steady-state framework for leaf temperature modeling, but as steady-state $g_s$ models were employed, iterations for stable solutions were still required. With the dynamic $g_s$ model presented in this study, the traditional nested iteration loops in leaf flux calculations, which can take up to 40 iterations to solve for a single simulation step in CLM4.5 (Bonan et al., 2018), can be replaced by more efficient and accurate prognostic updates of variables with ordinary differential
equations (ODEs). Such an approach can also facilitate better couplings of LSMs with other components in Earth system models (ESMs), where ODE systems are commonly used.

## 5   Conclusions

We implemented a simplified dynamic stomatal conductance model in CliMA-Land, and evaluated its impacts on model simulations across scales. In comparison with the traditional steady-state model, the dynamic model better predicted the coupled
temporal responses of $A_n$, $g_s$ and $C_i$ observed in leaf measurements. We also found uncertainties in parameter estimation for steady-state $g_s$ models with the traditional linear fitting method, when too short of a time step used resulted in unstable estimates. We proposed an alternative approach using a Bayesian nonlinear inversion framework with a dynamic model, which could help reduce the time investment for estimation, particularly for leaves with long time constants. Our results on canopy-scale simulations suggested that, although the effects of temporal $g_s$ responses on ecosystem fluxes may not be significant in
terms of monthly integrals, but should be take into account when predicting diurnal courses and quantifying sub-diurnal scale fluxes, as well as explaining the hysteresis patterns observed in diurnal cycles.

We demonstrated that the more realistic prognostic modeling of gradual $g_s$ response simplified the simulation as iteration loops for solving steady-states at each time step were avoided, and the dynamic model can be run at a finer time resolution that presents comparable computational costs to the current practice of steady-state leaf flux calculation. A similar framework can
be extended to leaf temperature modeling which will enable prognostic updates of leaf level variables with higher efficiency and accuracy, towards better couplings of LSMs with other components in Earth system models (ESMs).



*Code and data availability.* We coded our model and did the analysis using Julia, and the current version of the CliMA Land model with the implementation of dynamic stomatal conductance framework is available from the project website: https://github.com/CliMA/LandCliMA. Other code and data sets used in the study will be available in the final manuscript.

*Author contributions.* CF, KL and YW designed and conceptualized the study. TM provided the leaf gas exchange measurements. YW developed CliMA Land and helped with the implementation of the dynamic model. KL performed the analysis. KL, CF and YW interpreted the results. KL composed the manuscript with contributions from all authors.

*Competing interests.* The contact author has declared that neither they nor their co-authors have any competing interests.

*Acknowledgements.* We gratefully acknowledge the financial support of the Explorer Grant of Resnick Sustainability Institute at the Cali-
fornia Institute of Technology.



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



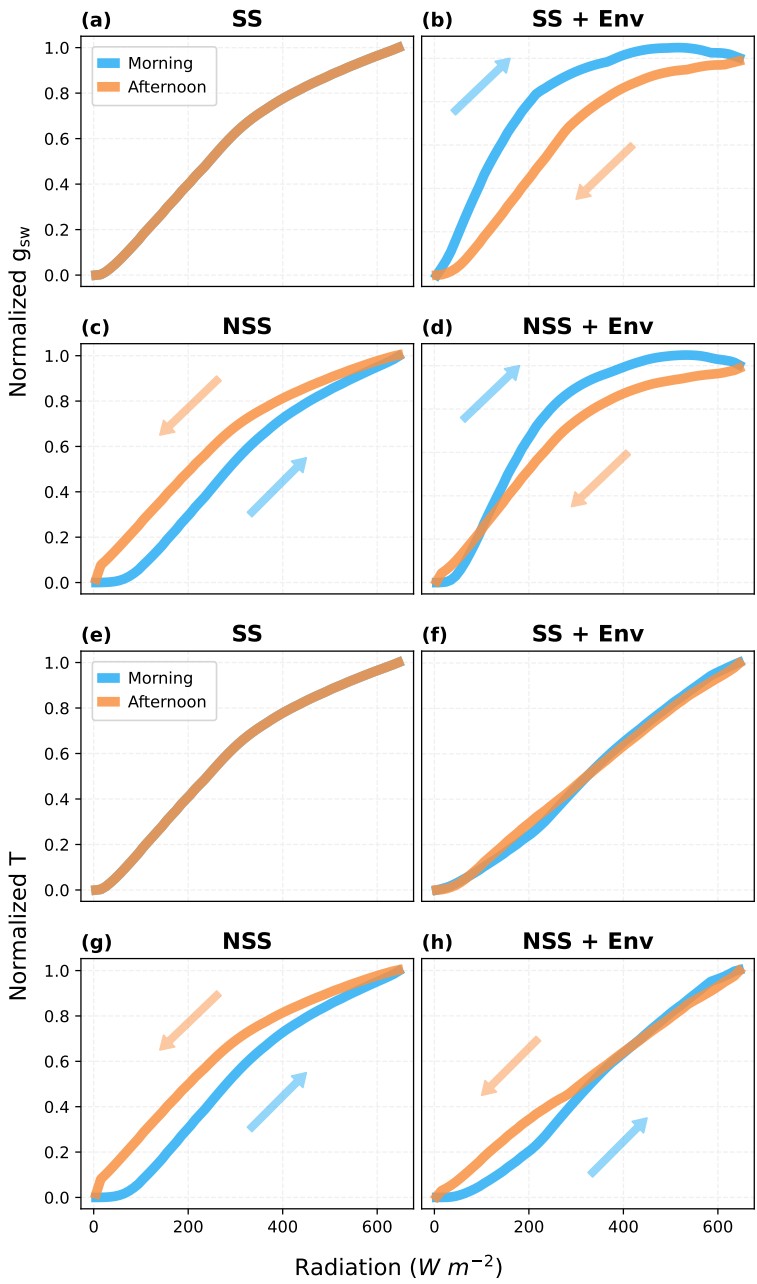

**Figure 10.** Hysteresis of the canopy-mean stomatal conductance ($g_{sw}$) and canopy transpiration rate ($T$) in response to radiation during an ideal clear-sky day. (a, e) SS model, (b, f) SS model with coupled diurnal variations of environmental conditions (Env, e.g. air temperature, VPD), (c, g) NSS model, (d, h) NSS model with Env. (a-d) normalized $g_{sw}$ responses, (e-h) normalized $T$ responses. In simulations without Env variations, except for the radiation, all the other environmental drivers were kept at the daytime means. $g_{sw}$ and $T$ is normalized with the values at noon (12:00). Arrows indicate the increasing and decreasing parts of the diurnal courses.



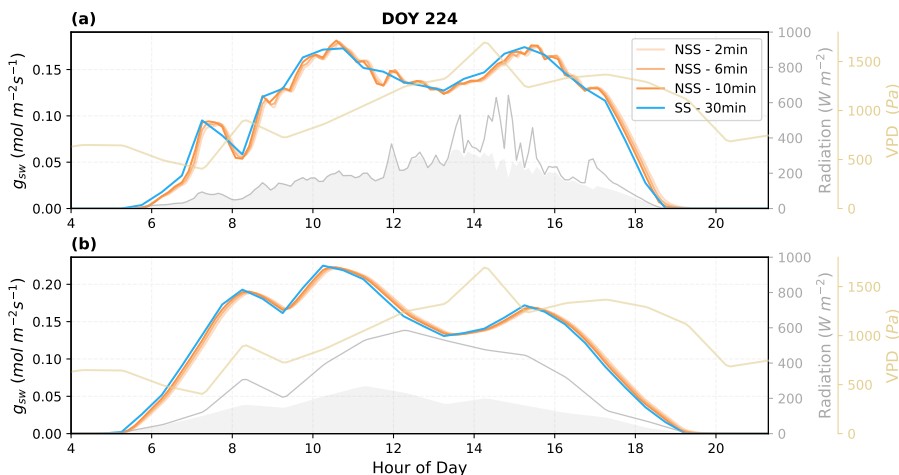

**Figure 11.** Simulations of dynamic $g_{sw}$ using different time steps (2 min, 6 min, 10 min) and comparison with the traditional steady-state modeling (30-minute resolution) predictions. a) using high temporal-resolution PAR as radiation input, values are resampled accordingly to match the time step used; b) using ERA5 hourly radiation as input, values are linearly interpolated to 30-minute resolution. The shaded areas in gray under the radiation curves represent the diffuse component of the total radiation.