# Peer review of "Non-steady-state Stomatal Conductance Modeling and Its Implications: From Leaf to Ecosystem"

_EGUsphere, 2023_

## Author Response (AR1)

**Review #1**

*This manuscript describes a new stomatal model that does not assume steady state responses to climate. Instead, this approach models gs with prognostic updates at each time step, which allows the model simulate limitations to gas exchange caused by stomatal speed and simplify leaf energy balance calculations.*

*The model was coupled with an LSM and evaluated at leaf and ecosystem levels. At leaf level the model could predict better the stomatal responses to changes in light regime. At ecosystem level, the model makes only a small difference at the canopy fluxes at mornings and evenings.*

*The paper is well written, and the results are generally clearly presented. I find the results of the paper interesting, and I really like the idea of the dynamic stomatal model because it seems a better representation of plant stomatal behaviour in a (rapidly) changing environment. I believe this approach have the potential to improve how LSMs represent ecosystem carbon and water dynamics. My only criticisms are relatively minor methodological issues explained in detail below. Additionally, I think the paper would be improved if the authors had also validated their dynamic model against ecosystem level observations (i.e. eddy flux data) similarly to what they did for leaf-level data.*

**Author response**: We would like to express our gratitude to the reviewer for taking the time to review our manuscript and provide valuable feedback. We agree with the reviewer that further validation on site-level measurements would be an improvement to illustrate the potential of our dynamic model. It can be the focus of follow-up research when detailed site-level datasets of eddy fluxes, plant traits, and meteorological conditions are available for accurate parameter calibration. We have added this point to the discussion section for future research directions (L352-353 in the revised manuscript):

*"Further improvements can be made in assessing other effects of $g_s$ temporal responses in LSM projections, as well as validating the comparisons with site-level observations. "*

*Specific comments:*

*L9: In the results you claim the daily effect of the dynamic model is notable (L244)*

**Author response**: Thanks for pointing this out. The overall effect is not significant, as the daily mean differences are mostly less than 2%, but depending on the variations of environment and when considering the mornings and afternoons separately, the impacts can be notable (eg. up to -7.4%). Our further tests also revealed that these differences can become substantial when the time constants of stomata opening and closure were different. We have updated the corresponding phrases for clarity:

synthesis and stomatal conductance to changes in light intensity using leaf measurements. In ecosystem flux simulations, while the impact of $g_s$ hysteresis response may not be substantial in terms of  monthly integrated

10    fluxes, our results  highlight the importance of considering this effect when quantifying fluxes in the mornings and evenings, and interpreting diurnal hysteresis patterns observed in ecosystem fluxes. Simulations also indicate that the biases in the integrated fluxes are more significant when stomata exhibited different speeds for opening and closure. Furthermore,

      In the simulations with same time constants of stomata opening and closure, the differences in fluxes between the NSS and SS predictions were not significant when integrated over monthly periods (e.g. the mean RD of transpiration in August 2017, 0.87 %, and the median, 1.0 %), but  can be notable at sub-diurnal scales depending on the  environmental conditions (e.g. the

280     variation of afternoon RDs ranged from -7.4 % to 6.1 %). When there were differences in $\tau_{op}$ and $\tau_{cl}$, the divergences between NSS and SS predictions can be more significant (e.g. when $\tau_{op} = 1/3\tau_{cl}$, the mean RD of transpiration in August 2017, 4.9 %, the maximum daily-mean RD of transpiration, 9.0 %).

*L24: The idea of using optimization to predict stomatal behaviour goes back to Cowan & Farquhar 1977, so I am not sure if using the term "more recently" is really appropriated.*

**Author response**: Thank you for the suggestion. The "more recently" in our writing mainly intended to refer to the implementation and usage of stomatal models in large-scale LSMs, as most LSMs have been using empirical models, such as CLM, LPJ-GUESS, etc. We agree when considering the overall history of the optimization approach, the term we used here is not the most appropriate one, we have replaced this term with *"Efforts have also been made to…"* (L25 in the revised manuscript) for clarity.

*L25: Most of the models discussed in these papers do not really optimize water use efficiency; they optimize the balance between carbon gain with some penalty, which can be related to plant hydraulics, non-stomatal limitations to photosynthesis, etc.*

**Author response**: Thank you for the correction. We acknowledge our current phrase is not a very accurate summary of the optimization approach. Optimal models optimize the trade-offs between carbon gain and a variety of penalties related to stomatal opening, which may not (only) include absolute water loss that WUE is calculated from. We will correct this phrase as follows for accuracy (L25-26 in the revised manuscript):

*"Efforts have also been made to constrain stomatal behavior from the principle of optimizing the trade-offs between carbon gain with the related penalty of stomatal opening."*

*L34-35: I assume it depends when the "next change occurs"? It would be useful to provide some quantitative examples of the time scales of stomatal response to environmental changes to clarify that.*

**Author response**: Thank you for the suggestion. We have added quantitative examples to better illustrate this point in our introduction  (L35-38 in the revised manuscript):

*"Plants can experience frequent environmental changes on a timescale of seconds, such as light fluctuations due to cloud cover and canopy shading. Meanwhile, stomatal response times vary from minutes to more than an hour. Thus, a steady state is often not reached when environmental conditions change faster than stomata can respond to."*

*L100: What steady-state model do you use to calculate the target gs?*

**Author response**: We used the Ball-Berry and Medlyn model (L112 in the original manuscript). More specifically, we used the Ball-Berry model for the demonstration of our dynamic framework at the leaf scale. At the canopy scale, we tested both the Ball-Berry and the Medlyn model. Since (1) the main focus of our study is the divergence of steady-state and prognostic modeling, the specific stomatal model used will not affect the comparison; (2) We have tested that the results of comparison using these two models were similar (as shown below); (3) We have vegetation trait estimation of the $g_1$ parameter for the Medlyn model in our test region, thus we used this model in the canopy modeling.

We have also updated the method section to clarify this information:

100    For prognostic updates of $g_{sw}$, we implemented a simplified dynamic model, adapted from previous studies on leaf-level prognostic modeling (Kirschbaum et al., 1988; Rayment et al., 2000; Noe and Giersch, 2004; Vialet-Chabrand et al., 2016):

$$\frac{\Delta g_t}{\Delta t} = \frac{(g_{ss} - g_t)}{\tau} \tag{3}$$

where $\Delta t$ is the time step of the simulation, $g_t$ represents the conductance at the current time step, $g_{ss}$ is the target conductance calculated with steady-state models at the current conditions, and $\tau$ is the time constant, representing the time scale of

105    stomatal responses. In this study, we used the Ball-Berry model (Ball et al., 1987) to compute the $g_{ss}$ for leaf level simulations for simplicity and the Medlyn model (Medlyn et al., 2011) for the canopy scale simulations, as the vegetation trait dataset (De Kauwe et al., 2015) we employed for our study region is only available for the Medlyn parameters. We should note that the selection of the empirical stomatal model is of minor relevance to our primary findings, as our study focuses on the differences between steady-state and prognostic schemes only.

**Results using the Medlyn model (using $g_1$ estimation from De Kauwe et al. (2015)):**

[Figure]

**Figure 8.** Relative differences (NSS - SS; RD) in the predicted daytime-mean fluxes of the NSS (time step: 1 min) and SS (1 min) simulations for August 2017. The solid line in each box indicates the median, and the dashed line represents the mean. The results for the transpiration rate ($H_2O$ flux), net productivity ($CO_2$ flux), canopy-averaged stomatal conductance to water ($g_{sw}$), and water-use efficiency (WUE) are shown in the respective columns from left to right. (a-d) $\tau_{op} = \tau_{cl} = 900$ s, (e-h) $\tau_{op} = 300$ s, $\tau_{cl} = 900$ s, (i-l) $\tau_{op} = 900$ s, $\tau_{cl} = 300$ s. Diurnal: 5:00-19:00, AM: 5:00-12:00, PM: 12:00-19:00.

**Results using the Ball-Berry model (using $g_1$ of 7.0):**

[Figure]

*L120: How many leaves/plants were used?*

**Author response**: We tested our model on four leaves, and for the manuscript, we included two example leaves with distinct time constants to illustrate the concept of temporal responses and show the differences in stomata behaviors across leaves, as well as how this may lead to further variations in parameter estimation when using traditional methods with the steady-state assumption. We updated the phrasing for clarity:

> in $APAR$, and the reproduced curves were close to the measurements, with all $R^2$ higher than 0.98. Fitted time constant $\tau$
> 225  showed a variation  between the two example leaves (292 s and 2028 s for the *Vitis vinifera* leaf and the *Juglans regia cv.*

*L140: This vcmax value should be micromol instead?  L205: check vcmax units*

**Author response**: Thanks for the correction. We have fixed the typos in these lines:

> 150  where $x_a$ is the prior estimate of the state (in this study, $V_{cmax}$: 70 µmol m$^{-2}$s$^{-1}$, $g_1$: 9, $g_0$: 0.03 mol H$_2$O m$^{-2}$ s$^{-1}$, $g_m$: 0.4
> mol CO$_2$ m$^{-2}$ s$^{-1}$, $\tau$: 600 s, $scaling_A$: 1); $S_a$ is the prior covariance matrix, assumed to be purely diagonal, with Gaussian

> 230  include $V_{cmax}$: 71 µmol m$^{-2}$s$^{-1}$, $g_1$: 11.3, $g_0$: 0.023 mol H$_2$O m$^{-2}$ s$^{-1}$, $g_m$: 0.18 mol CO$_2$ m$^{-2}$ s$^{-1}$, $scaling_A$: 1.1; for
> the *Juglans regia cv.* leaf, $V_{cmax}$: 152 µmol m$^{-2}$s$^{-1}$, $g_1$: 3.9, $g_0$: 0.052 mol H$_2$O m$^{-2}$ s$^{-1}$, $g_m$: 0.34 mol CO$_2$ m$^{-2}$ s$^{-1}$,
> $scaling_A$: 1.0.

*L170: Wouldn't a linear interpolation homogenize the environmental conditions over time and "mask" the real importance of the non-steady state model? I assume the dynamic model would only make a bigger difference in environments with rapidly changing environmental conditions. Would that explain why you have a relatively small impact of the dynamic model on the leaf and canopy simulations?*

**Author response**: We concur with the reviewer's observation that linear interpolations may homogenize variations and this could be a reason for the relatively small impacts. On the other hand, light fluctuations (for which we used high-resolution measurements) tend to be the most rapidly changing environmental condition compared to other variables (e.g. VPD, soil moisture, etc.) for which we applied linear interpolations. Thus, the effect of our interpolation may not be relatively significant. The magnitude of such impact can be assessed with measurements of higher temporal resolution for other env variables in future studies. We have updated the related methods section to clarify this point:

**2.3 Comparison of models in diurnal cycles**

165 To  assess the potential bias of the current steady-state modeling in LSMs, we compared the predictions of surface fluxes from models with different assumptions under natural environmental variations. We evaluated and compared the simulation results on both the leaf and canopy flux scales.

**2.3.1 Environmental drivers and plant traits**

170 As the light intensity tends to be the most rapidly changing environmental condition that stomata response to, we employed high temporal resolution radiation measurements in the field as the incoming irradiance inputs and ran CliMA Land with both setups.

**2.3.2**

Photosynthetically active radiation (PAR) in a crop field (42.481677°N, 93.523521°W) was recorded with a LI-190R quantum
175 sensor (LI-COR, Inc., Lincoln, NE, USA) at 1 s temporal resolution during August 2017.

*L175: This section needs more details on the LSM configuration for these simulations to allow reproducibility. For example, how canopy light diffusion, carbon allocation/vegetation dynamics and soil moisture were handled. Maybe add this information as supplementary material.*

**Author response**: Thanks for the suggestion. We have briefly summarized this information and included it in the supplementary material S1.1:

**S1.1 Climate Modeling Alliance (CliMA) Land Configuration**

In this study, we mainly employed the soil-plant-air continuum (SPAC) module of CliMA Land (github.com /CliMA/Land) to run simulations with different stomatal modeling frameworks. The SPAC module consists of four key sub-modules: canopy radiative transfer (RT), plant hydraulics, photosynthesis, and stomatal models (Figure S1). At each time-step, the SPAC module first calls the canopy RT module to compute the radiation condition for each canopy layer and leaf angle group, then it uses the photosynthesis, plantHydraulics and stomatal models modules to calculate the leaf-level stomatal conductance and photosynthesis rates, based on which it computes the canopy fluxes (Wang et al., 2023).

For the canopy RT, we employed a vertically layered canopy scheme with leaf angular distribution and a hyperspectral radiation transfer scheme (adapted from the Soil Canopy Observation of Photosynthesis and Energy fluxes model (SCOPE); (van der Tol et al., 2009; Yang et al., 2017)). For the photosynthesis module, we used the classic photosynthesis model developed by Farquhar et al. (1980) for C3 plants. For the stomatal models, due to the limited availability of hydraulic traits data in the test region, we applied a tuning factor based on soil water potential rather than plant hydraulics to $g_1$ to account for the response of stomata to water supply. The boundary layer conductance to water ($g_{bw}$) of leaf level predictions were prescribed using the estimated $g_{bw}$ provided in LI6800 measurements. The $g_{bw}$ in canopy scale simulations was assumed to be a constant at $3 \ \mathrm{mol\,m^{-2}\,s^{-1}}$, which is a relatively high conductance to make sure that the boundary layer conductance is not the main limiting factor of $CO_2$ supply (as our focus is on effects from stomatal conductance). We acknowledge more realistic $g_{bw}$ including calculation from wind speed and leaf width, which requires vertically resolved heterogeneous micro-climates and is under development.

In the steady-state mode, we ran iterations of the SPAC functions to find the stable solution for the given conditions at each time-step. In the prognostic mode, the simplified stomatal model was solved using the Euler method with a fixed step size, which is the time step used in each simulation. As current LSMs commonly use a time-step of 30 min or 60 min, we tested the stability of our model on 2, 6, and 10 min resolution for efficiency comparison, besides the fine 1 min resolution for flux comparison.

Vegetation traits and properties, including leaf chlorophyll content, leaf mass per area, leaf photosynthetic capacity, stomatal model $g_1$, Leaf area index, clumping index were prescribed for the test region, extracted from the global datasets (Croft et al., 2020; Butler et al., 2017; Luo et al., 2021; De Kauwe et al., 2015; Yuan et al., 2011; He et al., 2012). In this study, CliMA Land simulations were conducted offline. Environmental drivers (e.g. air temperature, dew-point temperature, volumetric soil water, wind speed etc.) were extracted from ERA5 reanalysis datasets (Hersbach et al., 2018) and updated accordingly at each step.

*L245: Do the daily differences "disappear" at monthly scales because you have opposite RD in different days that cancel each other out? If that is the case, it would still be interesting to show the total monthly differences between SS and NSS (using for example the mean absolute error between SS and NSS). I believe the differences between models throughout the month could still result in different trajectories for the vegetation over time, for example, if the higher carbon gain of a model in a given day resulted in more leaves being produced on that day, this model productivity advantage that would accumulate over the other one.*

**Author response**: The reviewer is correct, in our case, some of the daily differences are opposite and the aggregation makes the monthly differences less significant. We also agree with the reviewer that temporal responses of stomata could result in accumulated effects on vegetation growth. As the version of CliMA Land we employed has not implemented vegetation dynamics from net carbon gain from daily fluxes, we focused on evaluating the impacts of stomatal response on the gas exchange in this study. We appreciate the reviewer's suggestion and have extended the discussion section to include this as a direction for future research (L322-325 in the revised manuscript):

*"Further improvements can be made in assessing other effects of $g_s$ temporal responses in LSM projections, as well as validating the comparisons with site-level observations. For example, while daily effects on canopy productivity were minor, they may add up to significant differences in long-term vegetation growth trajectories. As plant traits were prescribed in our simulations, the accumulative effects were not included in our analysis of the short-term predictions."*

*L276: I could not find the computational cost differences in the results.*

**Author response**: At each time step in traditional simulations, iterations are needed for steady solutions for the coupled photosynthesis-stomatal conductance model. In CLM4, the default setting is a 3-step fix-point iteration, which would be at a similar computational cost

to our dynamic model if run at 10min resolution (3 sub-steps within 30min). In Section 3.3, we demonstrated that the dynamic model can be stably run at a resolution of 10min (L300 in the revised manuscript). Thus, we conclude that our dynamic modeling represents a comparable efficiency to traditional SS simulations while providing predictions at a finer resolution and eventually also facilitates prognostic leaf temperature treatment. Additionally, current LSMs require nested loops to also solve for the leaf temperature (Figure 1a), which can take up to 40 iterations at a single time step (L365 in the revised manuscript). We have now added clarification on this comparison:

325    does not always converge, leading to uncertainties in flux predictions. In our simulations at the canopy scale (Section 3.3), the dynamic model could be stably run at a temporal resolution (10 min) that presented comparable efficiency to the current practice of 30-minute resolution SS simulations commonly used in LSMs (3-step prognostic updates for each SS default 3-step iteration). This also indicates the dynamic model can enable predictions of canopy flux dynamics at a finer time resolution with higher efficiency and accuracy.

*L295: It could be interesting to see the cumulative effect of this unnecessary water loss on xylem hydraulic damage in future studies.*

**Author response**: Thank you for the suggestion. We also believe this could be an interesting and valuable direction for future research with LSMs that implemented parameterization of such effects. We have included this in the discussion of future studies (L357-358 in the revised manuscript):

*"Future studies can focus on the parameterization of these impacts in LSMs and the evaluation of cumulative effects on plant growth and hydraulics in the long term."*

*Fig. 2: I don't consider this figure really essential for the manuscript. Considering you already have 9 figures maybe it would be best to leave it as supplementary material.*

**Author response**: Thanks for the suggestion, we agree with the reviewer that it would be better to have it in supplementary materials. We have moved this figure to Figure S1.

*Fig. 6: Its hard to visualize the RD, please use darker shading.*

**Author response**: Thank you for the suggestion. We have updated the shading for better visualization as follows:

[Figure]

**Review #2**

*The manuscript by Liu et al. presents how including dynamic stomatal responses impacts gas exchange predictions from leaf to canopy. The topic is of high importance as research carried out at the leaf level has shown the importance of stomatal dynamics but there is still a debate about how these dynamics influence gas exchange at a larger scale. The manuscript is well written and I like that the authors are addressing this gap in the literature, however, I have some concerns about the methods used.*

**Author response**: We appreciate the time and effort the reviewer has dedicated to reviewing our manuscript and the valuable comments they provided for improving our study. We understand the reviewer's concern regarding our methods, and we appreciate the opportunity to explain and discuss these issues. We acknowledge that some concerns may have arisen due to the level of detail in our methods section. Thus, we have revised our methods section and added supplementary material to our model configuration for clarity.

*What should be clearer is what are the challenges to simulating gas exchange at different scales from leaf to canopy and from seconds to months. For example, during a diurnal period, the slow temporal response of stomatal conductance will lead to the transient limitation of photosynthesis and potential damage to the PSII reaction centres (due to excessive energy received compared to the sink strength). These effects accumulate during the day and impact plant growth, which will in turn affect the gas exchange of the following days. These effects are not clearly accounted for by the model. Averaging the fluxes over a long period is useful for quantifying the ecosystem exchange but does not reflect that every day the environmental conditions will impact the leaf functioning and acclimation.*

**Author response**: We acknowledge the valid point raised by the reviewer regarding the limitations of our model. The slow temporal response of stomatal conductance can have impacts on plants in various aspects, which, as the reviewer mentioned, include the direct gas exchange, as well as the potential damage to the PSII reaction centers, and other cumulative effects on leaf functioning. Here, our focus is on taking the first step to scale up the stomatal response effects on the gas exchange from leaf to canopy and ecosystem in an LSM. We agree that when accurate parameterization of other effects is available, future efforts can be made to quantify those impacts on the canopy scale and provide further understanding of how dynamic stomata response affects vegetation dynamics in the long term. We have extended our discussion section and addressed these challenges and limitations, as well as suggested future efforts in these directions (L351-358 in the revised manuscript):

*"Our study mainly focused on taking the first step to implement prognostic stomatal modeling in an LSM, including the impacts on canopy flux simulations. Further improvements can be made in assessing other effects of $g_s$ temporal responses in LSM projections, as well as validating the comparisons with site-level observations."..."The transient limitation on photosynthesis from the slow temporal response of $g_s$ can also cause potential photoinhibitory damage to the photosystem II reaction centers. Future studies can focus on the parameterization of these impacts in LSMs and the evaluation of cumulative effects of $g_s$ hysteresis on plant growth and hydraulics in the long term."*

*The effect of stomatal dynamics is also dependent on the fluctuation of the environment. The more it fluctuates, the more the stomatal behaviour will matter in the gas exchange.*

**Author response**: We agree with the reviewer that the relative effects of stomatal dynamics depend on the fluctuation of the environment, as this is the main reason we employed high-resolution measurements of radiation for the comparison simulations. We have updated the related sections to better address this point:

**2.3 Comparison of models in diurnal cycles**

To  assess the potential bias of the current steady-state modeling in LSMs, we compared
165   the predictions of surface fluxes from models with different assumptions  under natural environmental variations. As the light intensity tends to be the most rapidly-changing conditions among all the environmental cues that stomata generally response to (e.g. VPD, soil moisture), we employed high temporal resolution radiation measurements in the field as the light condition inputs and ran CliMA Land with both setups. We evaluated and compared the simulation results on both the leaf and canopy flux scales.

*The methods section is missing important information and requires more clarity.*

**Author response**: We acknowledge the concern that the reviewer brought out about the level of detail in our previous methods section, and we have briefly summarized the CliMA Land configuration and added this information to the supplementary materials S1.1.

*From lines 90 to 105, it is not clear how gbc was set or calculated.*

**Author response**: For leaf-level predictions, gbc is prescribed with the estimated gbc from LI6800 measurements. For canopy-scale simulations, in the CliMA Land version we employed in this study, we used a reasonable and fixed gbc value of 3/1.35 (gbw is assumed to be 3), which is relatively high conductance to make sure the gbc is not the main limiting factor of CO2 supply (as our focus is on effects from stomatal conductance). We acknowledge more realistic gbc values including calculation from wind speed and leaf width,

which can be challenging on the canopy scale. As the vegetation module of CliMA Land implemented a vertically layered canopy scheme (rather than a big-leaf scheme), such calculation would require resolved wind speed at each layer and coupling with the atmospheric module, which is still under development. Thus, we set a reasonable gbc for all layers, which we believe will not significantly impact on our results, as the focus is the comparison of the two modeling approaches, rather than the absolute fluxes. We have also included this information in the SI (S1.1):

*"The boundary layer conductance to water ($g_{bw}$) of leaf level predictions were prescribed using the estimated $g_{bw}$ provided in LI6800 measurements. The $g_{bw}$ in canopy scale simulations was assumed to be a constant at 3 μmol m$^{-2}$ s$^{-1}$, which is a relatively high conductance to make sure that the boundary layer conductance is not the main limiting factor of $CO_2$ supply (as our focus is on effects from stomatal conductance). We acknowledge more realistic $g_{bw}$ values including calculation from wind speed and leaf width, which requires vertically resolved heterogeneous micro-climates and is under development."*

*How the simplified model was solved and with which procedure is missing. Did the authors use an ODE solver?*

**Author response**: The simplified model is solved with a simple Euler method with fixed step sizes (the time steps used in simulations), we have added this information (S1.1):

*"In prognostic mode, the simplified stomatal model was solved with the Euler method with a fixed step size, which is the time steps used in each simulation."*

*How are the environmental variables included in this model? Did the authors use them as forcing variables that change continuously?*

**Author response**: Yes, environmental variables from ERA5 reanalysis dataset (e.g. air temperature, dew-point temperature, volumetric soil water, wind speed etc.) are used as meteorological drivers and updated accordingly at each time step. We have clarified this in the revised manuscript:

the month of August 2017. Meteorological variables are updated at each time step. In order to further evaluate the potential

210    contribution of $g_s$ hysteresis to the observed diurnal hysteresis of ecosystem fluxes, we compared the standard runs with the

Also in the SI (S1.1): *"In this study, CliMA Land simulations were conducted offline. Environmental drivers (e.g. air temperature, dew-point temperature, volumetric soil water,*

*wind speed etc.) were extracted from ERA5 reanalysis datasets and updated accordingly at each step."*

*How is the steady state gs calculated? Which model was used: Ball or Medlyn? How were they calibrated?*

**Author response**: We used the Ball-Berry and Medlyn model (L112 in the original manuscript). More specifically, we used the Ball-Berry model for the demonstration of our dynamic framework at the leaf scale and calibrated with the framework we described in Section 2.2.2. At the canopy scale, we tested both the Ball-Berry and the Medlyn model. Since (1) the main focus of our study is the divergence of steady-state and prognostic modeling, the specific stomatal model used will not affect the comparison; (2) We have tested that the results of comparison using these two models were similar (as shown in the response to the first reviewer, page 4); (3) We have vegetation trait estimation of the $g_1$ parameter for the Medlyn model in our test region, thus we used this model in the canopy modeling.

We have also updated the method section to clarify this information:

100  For prognostic updates of $g_{sw}$, we implemented a simplified dynamic model, adapted from previous studies on leaf-level prognostic modeling (Kirschbaum et al., 1988; Rayment et al., 2000; Noe and Giersch, 2004; Vialet-Chabrand et al., 2016):

$$\frac{\Delta g_t}{\Delta t} = \frac{(g_{ss} - g_t)}{\tau} \tag{3}$$

where $\Delta t$ is the time step of the simulation, $g_t$ represents the conductance at the current time step, $g_{ss}$ is the target conductance calculated with steady-state models at the current conditions, and $\tau$ is the time constant, representing the time scale of

105  stomatal responses. In this study, we used the Ball-Berry model (Ball et al., 1987) to compute the $g_{ss}$ for leaf level simulations for simplicity and the Medlyn model (Medlyn et al., 2011) for the canopy scale simulations, as the vegetation trait dataset (De Kauwe et al., 2015) we employed for our study region is only available for the Medlyn parameters. We should note that the selection of the empirical stomatal model is of minor relevance to our primary findings, as our study focuses on the differences between steady-state and prognostic schemes only.

*At different places in the manuscript, the time steps used are different and without a clear explanation of why, it is confusing.*

**Author response**: Thanks for pointing this out. In our leaf-level runs: as the length of total measurements is around 4.5 hours, to illustrate the effects of dynamic stomatal conductance response with higher accuracy, we used a relatively fine resolution of 10s.

On the canopy scale, for the comparison of differences in gas exchange, considering the balance between computational cost and accuracy, we chose a practical resolution of 1min for non-steady-state runs and compared it with traditional steady-state simulations at 30min resolution (the commonly used time step of LSMs). For the comparison of model efficiency,

we also ran our model on 2, 6, and 10min (Figure 10) resolution to test model stability. We have also included this clarification in the SI (S1.1):

*"In the prognostic mode, the simplified stomatal model was solved using the Euler method with a fixed step size, which is the time steps used in each simulation. As LSMs commonly use a time-step of 30 min or 60 min, we tested the stability of our model on 2, 6, and 10 min resolution for efficiency comparison, besides the fine 1 min resolution for flux comparison."*

*The authors did not include leaf energy balance to calculate leaf temperature, but this can have an important impact on their simulations. The vapour pressure gradient between the leaf and atmosphere depends on this and drives the water exchange. This should be acknowledged at least as a limitation in the discussion. Overall, it is difficult to judge the validity of the simulations, because important details are missing.*

**Author response**: Thank you for the suggestion. We agree with the reviewer that the calculation of leaf temperature from energy balance impacts our simulation, and the stomatal response contributes to the latent heat fluxes, as indicated in Figure 1 and the discussion section (L319-325 in the original manuscript). This is certainly an area of specific interest to us, as prognostic gs treatment can also cause larger temperature fluctuations at the leaf scale. The implementation of leaf energy balance for leaf temperature with dynamic stomatal response would also require a non-steady-state energy balance modeling (in traditional LSMs, this is done with nested loops for steady solutions, as in Figure 1), which is under development in the latest version CliMA Land. We also plan to include dynamic leaf temperature in future simulations, so that the leaf flux calculation in our LSM can become fully prognostic, and we will also be able to better quantify the effects of stomatal response. We have further clarified this point in the related part of our discussion section:

> The dynamic $g_s$ model enables predictions of temporal changes in latent heat flux through transpiration in leaf energy balance, which allows a similar prognostic framework to be employed for the modeling of leaf temperature. Coupled dynamic
>
> 385 modeling of stomatal conductance and leaf temperature will enhance our ability to evaluate the influences of $g_s$ hysteresis on the feedback between leaf transpiration and thermal condition. This is out of scope of our current study but can be a valuable direction for future research efforts. Bonan et al. (2018) implemented a non-steady-state framework for leaf temperature mod-

*For the exponential model used here, there is a time constant for stomatal opening and another one for stomatal closure. Stomata have very different rapidity of response for opening and closing. Here the authors only used one time constant and only estimated it for the opening part. The asymmetry of the time constants is strongly influencing the hysteresis between the morning and afternoon part of the simulation. This should be corrected.*

**Author response**: Thanks for the suggestion. We have added tests with different stomatal opening and closure speeds by using separate time constants:

**2.3.2 Model simulations**

 We ran the CliMA Land surface flux simulations with different

195     stomatal modeling schemes to assess the effects of $g_s$ temporal response on model predictions. In the SS runs, iterations were employed to converge to steady-state solutions at each time step. For NSS mode, previous studies have observed different time constants for stomatal opening ($\tau_{op}$) and closure ($\tau_{cl}$) in various species, as well as a positive correlation between $\tau_{op}$ and  $\tau_{cl}$, with of the $\tau_{op}/\tau_{cl}$ ratio varying from around 1/3 to 3 (McAusland et al., 2016; Vico et al., 2011; Ozeki et al., 2022). Based on the average time con-

200     stant retrieved from  the leaf response curves in Section 2.2.1 and 2.2.2 as well as previous  estimates on the time constant variations (Vialet-Chabrand et al., 2013; McAusland et a , we tested several sets of $\tau_{op}$ and $\tau_{cl}$ varying from 300 s to 900 s, including (a) $\tau_{op} = \tau_{cl} = 900$ s, as the base comparison; (b) $\tau_{op} > \tau_{cl}$, with $\tau_{op}/\tau_{cl}$ ratios varying from 1.2 to 3, e.g. $\tau_{op} = 900$ s, $\tau_{cl} = 300$ s; (c) $\tau_{cl} > \tau_{op}$, with $\tau_{cl}/\tau_{op}$ ratios from 1.2 to 3,

205     e.g. $\tau_{op} = 300$ s, $\tau_{cl} = 900$ s.

Our results indicated that although the relative differences in stomatal opening and closure speeds can affect the magnitude of hysteresis, the patterns we observed and discussed still remained similar, even when the time constant for opening is 3 or ⅓ times of the closing time constant (Figure S6 and S7).

310     of coupled environmental variations (Figure 9 g and h), in which canopy $H_2O$ fluxes tended to be higher in the afternoon. Differences between $\tau_{op}$ and $\tau_{cl}$ affected the magnitudes of hysteresis, but the overall patterns remained similar (Figure S6, S7).

[Figure]

Figure S6: Hysteresis of the canopy-mean stomatal conductance ($g_{sw}$) and canopy transpiration rate ($T$) in response to radiation during an ideal clear-sky day, when $\tau_{op} = 300\,\text{s}$, $\tau_{cl} = 900\,\text{s}$. (a, e) SS model, (b, f) SS model with coupled diurnal variations of environmental conditions (Env, e.g. air temperature, VPD), (c, g) NSS model, (d, h) NSS model with Env. (a-d) normalized $g_{sw}$ responses, (e-h) normalized $T$ responses. In simulations without Env variations, except for the radiation, all the other environmental drivers were kept at the daytime means. $g_{sw}$ and $T$ is normalized with the values at noon (12:00). Arrows indicate the increasing and decreasing parts of the diurnal courses.

More interestingly, comparisons have also revealed that when the time constants of stomatal opening and closure response differed, the cumulative differences between NSS and SS simulations tended to increase (Figure 8, S3, S4). We have included the related analysis in the revised manuscript.

In the results section:

270  When $\tau_{op} = \tau_{cl}$, the overestimate of morning $g_{sw}$ in SS predictions was mostly compensated by an underestimate of afternoon $g_{sw}$, resulting in relatively minor differences in daily average $g_{sw}$ (Figure 8c; the mean RD of daytime-mean $g_{sw}$, 0.5 %). However, when the time constants of stomatal opening and closure were not equal, there is overall underestimation or overestimation in both mornings and afternoons, and the daily-mean RDs can be notable (Figure 8g and k, Figure S3, Figure S4). For example, when $\tau_{op} = 3\tau_{cl}$, the faster closure than opening of stomata led to overall lower conductance

275 over the diurnal cycles, compared to SS runs (Figure 8k; the mean RD of daytime-mean $g_{sw}$, -6.1 %).

In the simulations with same time constants of stomata opening and closure, the differences in fluxes between the NSS and SS predictions were not significant when integrated over monthly periods (e.g. the mean RD of transpiration in August 2017, 0.87 %, and the median, 1.0 %), but  can be notable at sub-diurnal scales depending on the  environmental conditions (e.g. the

280  variation of afternoon RDs ranged from -7.4 % to 6.1 %). When there were differences in $\tau_{op}$ and $\tau_{cl}$, the divergences between NSS and SS predictions can be more significant (e.g. when $\tau_{op} = 1/3\tau_{cl}$, the mean RD of transpiration in August 2017, 4.9 %, the maximum daily-mean RD of transpiration, 9.0 %).

The overall tendency to overestimate productivity with traditional SS models was also observed  at the canopy scale, when $\tau_{op}$ was equal to or larger than $\tau_{cl}$, as the regulation of $g_s$ hysteresis on the supply of $CO_2$ for photosynthesis was not

285 considered (Figure 8b and j). For example, in Figure 6, when rapid spikes of radiation occurred in the afternoon, the speed of $g_s$ response constrained the increases of photosynthesis in the NSS simulation. However, when $\tau_{op}$ was smaller than $\tau_{cl}$, predicted daily-mean photosynthesis is slightly higher in NSS simulation (Figure 8f; the mean RD of productivity in August 2017, 0.2 %). This resulted from the overall higher $g_{sw}$, due to faster opening than closure (Figure 8g), as higher conductance resulted in higher $C_i$ (Figure S5), leading to generally higher rates of photosynthesis.

290 In contrast to the leaf-scale results, when accounting for other co-varying environmental drivers (e.g. temperature, VPD, soil water content), the SS model tended to underestimate canopy transpiration rates,  when $\tau_{op} = \tau_{cl}$ (Figure 6b, Figure 7b, Figure 8a). This could be because the transpiration rates were determined by both $g_{sw}$ and VPD. During the daytime, VPD  typically increased following air temperature and peaked in the afternoon, when the slow response of stomata to the increasing VPD

295 and decreasing radiation could result in excess water loss (Figure 7b, Figure 8 a and c). The overestimation of productivity and underestimation of transpiration in SS simulations  both contributed to the overestimation of WUE. When $\tau_{op} < \tau_{cl}$, slower stomatal closure led to increased water loss and thus a more significant underestimation of transpiration in the SS predictions (Figure 8e; the mean RD of transpiration in August 2017, 4.9 %), resulting in further overestimation of the WUE.

[Figure]

**Figure 8.** Relative differences (NSS - SS; RD) in the predicted daytime-mean fluxes of the NSS (time step: 1 min) and SS (1 min) simulations for August 2017. The solid line in each box indicates the median, and the dashed line represents the mean.  The results for the transpiration rate H₂O flux),  net productivity CO₂ flux), canopy-averaged stomatal conductance to water ($g_{sw}$sw),  and water-use efficiency (WUE) are shown in the respective columns from left to right. (a-d) $\tau_{op} = \tau_{cl} = 900\,\text{s}$, (e-h) $\tau_{op} = 300\,\text{s}$, $\tau_{cl} = 900\,\text{s}$, (i-l) $\tau_{op} = 900\,\text{s}$, $\tau_{cl} = 300\,\text{s}$. Diurnal: 5:00-19:00, AM: 5:00-12:00, PM: 12:00-19:00.

In the discussion:

Furthermore, we evaluated how the inclusion of $g_s$ temporal responses could affect model predictions of leaf and canopy

340 fluxes in diurnal cycles with natural environmental variations (Section 3.2.2). The comparison of NSS and SS simulations indicated that  while the differences in fluxes depended on the integration timescales, relative speed of stomatal opening and closure, and environmental variations. In terms of instantaneous effects, slow opening of stomata tended to limit productivity responses to rapid radiation increases, and delayed closure of $g_s$ following decreases in

345 radiation or increases in environmental stress (e.g. increasing VPD)  resulted in unnecessary water loss.  The divergence of NSS and SS schemes was less significant when considering the monthly integrated canopy fluxes, compared to daily or sub-diurnal scale results. The monthly differences were more notable when the speeds of stomata opening and closure differed. The overall effects on WUE also depended on the relative speed of opening and closure. In the simulations where stomata open at a similar or faster speed than they close, excessive water

350 loss in the afternoons, when VPD was high, led to a lower WUE. This also suggested that traditional steady-state simulations   may overestimate WUE. Similar impacts have been noted in studies on leaf-scale response to PPFD fluctuations (Lawson et al., 2011; Lawson and Blatt, 2014; McAusland et al., 2016).  Meanwhile, when stomata opened more slowly than they closed, plants exhibited both a lower maximum $g_s$ during diurnal cycles and a lower average $g_s$ compared to the SS runs. This resulted in reduced transpiration and increased WUE, even though productivity was also suppressed. These

355 results suggest that the temporal hysteresis of $g_s$ can have impacts on integrated canopy fluxes, and further studies on variations of stomata opening and closure speeds across plants can be helpful to assess these effects more comprehensively

360 on larger scales.

In the abstract (L11 in the revised manuscript):

evenings, and interpreting diurnal hysteresis patterns observed in ecosystem fluxes. Simulations also indicate that the biases in the integrated fluxes are more significant when stomata exhibit different speeds for opening and closure. Furthermore, prog-

*The authors concluded that their optimisation procedure improves the parameter estimation compared to traditional linear fitting methods. It is not clearly explained in the text how these methods differ and what are the improvements. The need to reach steady state gs to estimate the parameter values is known. Using a coupled dynamic gs model and steady-state target model to fit the data of a light response curve has been done previously. In general, it is an interesting result but not the focus of this study.*

**Author response**: Our conclusion about this procedure is mainly that it can provide a valuable alternative approach for parameter estimation. We successfully employed this approach to calibrate the parameters for our dynamic model and reproduced the leaf response curves. Compared to traditional linear fitting methods, (a) the Bayesian nonlinear inversion framework can optimize multiple parameters based on a joint fit of An and gs response curves. In the traditional linear fits, Vcmax is often estimated with A/Ci response

curve, and stomatal parameters are often estimated separately with light response curves. This approach can also be employed to estimate parameters with both A/Ci and light response curves. (b) As the reviewer mentioned, the linear regression method requires reaching an equilibrium at each environmental condition, which can be time-consuming. The bias of estimation with too short of a time step has also been discussed in previous studies. In the meantime, (c) our fitting with the dynamic model does not require steady states in principle, which can help reduce the time investment in parameter calibration. Our main goal of this study is to implement the dynamic model in our LSM and illustrate some of its implications. As the reviewer suggested, this part is not our main focus of this study but one of the implications we would like to point out – the implementation of the dynamic model can provide an alternative way of parameter estimation without the steady-state assumption.

*Line 25: the optimisation theory does not optimize water use efficiency.*

**Author response**: Thank you for the correction. The other reviewer has also pointed out our inaccurate description of optimization models. We acknowledge they mainly optimize the balance between carbon gain and a variety of potential penalty functions related to stomatal opening, which may not (only) include absolute water loss that WUE accounts for. We will correct this phrase as follows for accuracy (L25-26 in the revised manuscript):

*"Efforts have also been made to constrain stomatal behavior from the principle of optimizing the trade-offs between carbon gain with the related penalty of stomatal opening."*

---

## Author Response (AR2)

*The authors have addressed most of my previous comments. I still have a few points to improve:*

**Author response**: We thank the reviewer for the final thoughtful comments. All three remaining minor concerns had to do with potential numerical implementation impacts. In all three cases, we ran tests to alleviate the final concerns and conclude that our findings are not impacted. Please find a more detailed response to each individual points below:

*- The boundary layer conductance used in the model could be estimated using a canopy model. There are estimates for fields and forest that would be more representative of the reality. At least can the authors compare the value of the Licor to those of such a canopy model to see if it is in the same order of magnitude?*

**Author response**: The daytime mean 10m wind speed at our site is 3.6 ± 2 m s⁻¹. According to Bonan (2019), p.161 (screenshots below), the typical boundary layer conductance at this wind speed is around 1.0 to 2.2 mol m⁻² s⁻¹ for 5cm-wide leaves, which is in the same order of magnitude as the $g_{bw}$ assumed in our simulations at the canopy scale (3 mol m⁻² s⁻¹).

[Figure]

**Figure 10.4** Leaf boundary layer conductance for heat $g_{bh}$ in relation to wind speed for a 5 cm leaf and a 10 cm leaf. Shown are the laminar and turbulent conductance for each leaf. The turbulent conductance exceeds the laminar conductance for Re > 16,248. Dashed lines show laminar and turbulent conductance beyond this threshold. The inset panel shows conductance with laminar free convection. For comparison, the symbols show forced convection conductance with a wind speed of 0.1 m s⁻¹. Calculations are at STP (15°C and 1013.25 hPa).

vary with temperature and pressure (Table A.3). At 15°C, the conductance for $H_2O$ relative to heat is $g_{bw} = 1.10g_{bh}$ and relative to $CO_2$ is $g_{bw} = 1.36g_{bc}$. More commonly, it is assumed that $g_{bw} = g_{bh}$ and $g_{bw} = 1.4g_{bc}$.

As the total conductance is $(g_{bw}^{-1} + g_{sw}^{-1})^{-1}$, and the mean $g_{sw}$ ranges from 0 to 0.3 mol m⁻² s⁻¹, the total conductance is mostly constrained by $g_{sw}$. Thus, boundary conductance at moderate wind speeds will have limited impacts on total conductance, and it does not affect the interpretation of our results. We have also tested that, even a relatively low prescribed gbw (0.4 mol m⁻² s⁻¹) does not affect the magnitude nor the direction of our flux comparison between the NSS and SS scheme (relative differences), despite changes in absolute fluxes.

*- Using a simple euler method is prone to errors (and dependent of the time step) that are not controlled here. Did the authors test that the simple method gives the same results as methods such as rk45 or BDF?*

**Author response**: We assessed the stability of our explicit forward-Euler method by comparing the results to simulations with much finer time steps and found negligible differences, underlining that our time step was small enough and the solution stable. In the future, more advanced solvers such as higher-order Runge Kutta solvers could be applied, but here we focused on the concept, not the numerical details.

We have clarified the stability concern in the revised SI as follows: *"We also tested that our method provides similar results with a much finer time step (1s, 1/60 of the 1min time step used for the comparison), the relative difference is minimal, 0.2 ± 0.1 %, indicating the time step we chose is sufficient for our simulations and comparison."*

*- In the original model, splines are fitted over environmental variables and used in the solver to solve the temporal response. This simulates a continuous environment variation and its impact on gs, not a stepwise variation (constant within each time step) as it seems the case here. Did the authors test how much it would impact their results?*

**Author response**: Thanks for the suggestion. We used linear interpolation for continuous environmental variations, which can have fluctuations in the derivatives. The impacts of the variation within our time steps can be assessed by comparing the results with finer time-step simulations, as mentioned in the response above, this does not change our results. We also tested the impact of different fitting methods for environmental variables, and the relative differences between the simulation with spline-fitted environmental variables (at 6s resolution) and the linear interpolation is 0.01 ± 0.1%, which is also minimal and does not impact our findings.

Reference:
Bonan, G.: Climate Change and Terrestrial Ecosystem Modeling, Cambridge University Press, 459 pp., 2019.